# DReS-FL: Dropout-Resilient Secure Federated Learning for Non-IID Clients via Secret Data Sharing

**Jiawei Shao[‡], Yuchang Sun[‡], Songze Li[‡§], Jun Zhang[‡]**
[‡]Hong Kong University of Science and Technology
[§]Hong Kong University of Science and Technology (Guangzhou)
{jiawei.shao,yuchang.sun}@connect.ust.hk, {songzeli,eejzhang}@ust.hk

## Abstract

Federated learning (FL) strives to enable privacy-preserving training of machine learning models without centrally collecting clients' private data. Despite its advantages, the local datasets across clients in FL are non-independent and identically distributed (non-IID), and the data-owning clients may drop out of the training process arbitrarily. These characteristics will significantly degrade the training performance. Therefore, we propose a *Dropout-Resilient Secure Federated Learning* (DReS-FL) framework based on Lagrange coded computing (LCC) to tackle both the non-IID and dropout problems. The key idea is to utilize Lagrange coding to secretly share the private datasets among clients so that each client receives an encoded version of the *global dataset*[1], and the local gradient computation over this dataset is unbiased. To correctly decode the gradient at the server, the gradient function has to be a polynomial in a finite field, and thus we construct *polynomial integer neural networks* (PINNs) to enable our framework. Theoretical analysis shows that DReS-FL is resilient to client dropouts and provides strong privacy guarantees. Furthermore, we experimentally demonstrate that DReS-FL consistently leads to significant performance gains over baseline methods.

## 1 Introduction

Federated learning (FL) [1] is a machine learning framework in which a central server coordinates a large number of clients to collaboratively train a shared model. The key idea of FL is to train the model locally by individual clients and aggregate updates globally by the server. The main target is to provide privacy protection for clients' local samples and solve the "data islands" problem. However, as local data are typically non-independent and identically distributed (non-IID), the model divergence during the local update may lead to unstable and slow convergence [2, 3, 4]. With many clients involved in the training, some of the clients could drop out of the training process unexpectedly (due to poor connectivity, battery level, etc), and it will cause detrimental model performance [5]. Thus, effective mechanisms are needed to tackle the non-IID data distribution and client dropouts, while preserving the privacy of local datasets, which motivates this work.

To alleviate the non-IID problem, existing methods typically follow *algorithm-based approaches* [2, 6, 7, 8, 9] and add regularization terms to mitigate the model divergence. However, these methods are not dropout-resilient evidenced by the empirical results in [10]. This can be explained by the greatly varying data distributions among different rounds. Another fold of strategy for dealing with the non-IID problem is *data-centric approach* [11, 12, 13, 14, 15, 16, 17, 18, 19, 20, 21], which generates extra training samples to construct a more balanced data distribution for each client. The common practices are to share the synthesized samples [13, 14, 15, 16] or GAN-based augmented

---

[1]In the context of this paper, we use the global dataset to denote the concatenation of the clients' datasets.

data [17, 18, 19, 20, 21]. However, these methods may leak private information about local datasets and violate the privacy criterion in FL.

In this work, we develop a ***Dropout-Resilient Secure Federated Learning*** (DReS-FL) framework to address the above problems via Lagrange coded computing (LCC) [22]. The key idea of LCC is to encode the datasets using Lagrange polynomials that create computational redundancy across the workers in a privacy-preserving way to tolerate client dropouts. Before the training starts, the clients secretly share their encoded datasets with each other. This allows clients to access an encoded version of the global dataset that solves the data heterogeneous problem. In each communication round of federated training, the clients perform local gradient computations on the mini-batches sampled from the encoded datasets. After collecting the uploaded computation results from surviving clients, the server performs polynomial interpolation to decode the *global gradient*[2] for model training. Therefore, the training process in DReS-FL is made equivalent to centralized training and eliminates the non-IID and dropout problems. With respect to privacy protection, the proposed framework has two salient features:

- It guarantees the privacy of local datasets during data sharing, i.e., no private information can be inferred from the encoded data even if a certain number of clients collude.

- It achieves the same privacy guarantee as the *secure aggregation* protocols, i.e., the server learns no information about the private dataset from a single client's computation result.

Note that to correctly decode the gradient at the server, the gradient has to be a polynomial function in a finite field, which is a main design challenge of DreS-FL. To sum up, our main contributions are summarized as follows:

- The proposed DReS-FL framework provides a unified approach to tackle two critical problems of FL, namely, *non-IID data distribution* and *client dropouts*. Meanwhile, it maintains privacy and security guarantees such that no information about local datasets can be leaked beyond the global model parameters.

- We construct *polynomial integer neural networks* (PINNs) to ensure that the gradient is a polynomial, so that cryptographic primitives can be applied for secure computation. A PINN consists of affine transformation layers with parameters constrained in an integer set, and it adopts the quadratic function as the activation function. The convergence analysis of DReS-FL with PINNs is also provided.

- We conduct extensive experiments on FL benchmark datasets to demonstrate the effectiveness of DReS-FL. It is shown that DReS-FL outperforms baseline methods under the setting where local datasets are heterogeneous and clients may drop out of the training process arbitrarily.

## 2 Related Works

**Non-IID data and client dropouts.** Training with heterogeneous data is a unique challenge for FL [1], which significantly affects the convergence performance [5]. The client dropouts exacerbate the non-IID problem as the data distributions among different rounds could vary greatly. Many algorithm-based methods [2, 6, 7, 8, 9] attempt to mitigate the clients' model divergence, but these methods cannot solve the essence of the non-IID problem due to the intrinsic difference between minimizing the local empirical loss and minimizing the global empirical loss. Another line of work adopts data-centric methods [17, 18, 19, 20, 21] to modify the local distributions. Ideally, a perfect data sharing mechanism should achieve that the local datasets have the same distribution as the global dataset while maintaining the privacy guarantee. Common practices include sharing raw datasets [11, 12], synthesized samples, [13, 14, 15, 16] or augmented data [17, 18, 19, 20, 21]. However, these works cannot fully preserve local data privacy in an information-theoretic sense [23]. A special data-centric method is the secret coding scheme, which has been widely utilized

---

[2]The global gradient corresponds to the stochastic gradient computed from mini-batches that are uniformly sampled from the global dataset. For simplicity, we consider that clients only perform one local stochastic gradient descent (SGD) step in each communication round. Note that the proposed DReS-FL framework, as discussed in Appendix D, can be extended to more general cases in which clients can run multiple local SGD steps.

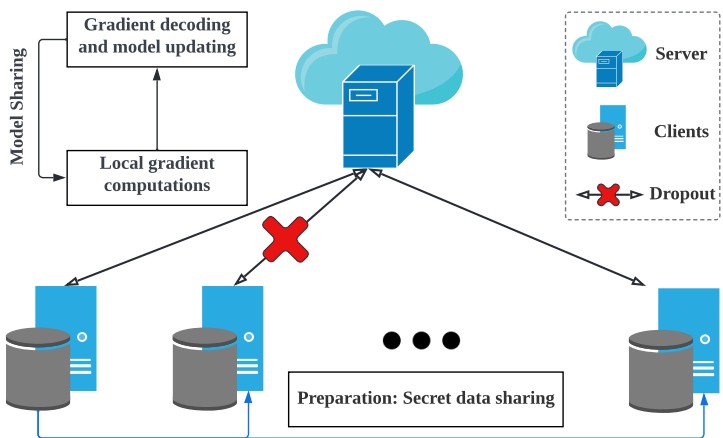

Figure 1: The DReS-FL system model. At the beginning of training, the clients secretly share the local datasets with each other. Then, the model parameters are iteratively trained by (1) local gradient computations and (2) gradient decoding and model updating until convergence.

in homomorphic encryption (HE) [24, 25, 26, 27, 28, 29, 30] and multiparty computation (MPC) techniques [31, 32, 33]. This coding scheme allows computations to be performed on encrypted data and has been used for privacy-preserving machine learning [26, 30, 33]. However, the HE methods often suffer from time-consuming cryptographic tools, and MPC techniques are difficult to generalize such primitives to a large number of clients. Recently, distributed secure machine learning frameworks [34, 35] have been proposed for logistic regression problems. They apply Lagrange coding for secret data sharing and approximate the Sigmoid function by a polynomial function. This paper proposes DReS-FL to further extend these works to train deep neural networks in the FL setting.

**Secure aggregation.** It has been shown recently that the clients' updates in FL may reveal substantial information about the local datasets, and the private training data can be reconstructed through model inversion attacks [36, 37, 38]. To prevent information leakage from the local models, *secure aggregation* protocols [39, 40, 41, 42] have been developed to allow for global aggregation without revealing the parameters of clients' models. Even if some clients may drop out, these protocols can still recover the aggregated results of the surviving clients. Existing protocols essentially rely on two main principles, including a pairwise random-seed agreement for mask cancellation and secret sharing of the random seeds to construct the dropped masks [39, 43, 40, 41, 44, 42]. However, these approaches may suffer from severe performance degradation in non-IID settings, since the surviving clients in each round vary greatly, and thus the aggregate gradient is biased towards different data distributions. Different from previous works, our proposed DReS-FL framework achieves the same privacy guarantee while solving the data heterogeneity problem.

## 3 System Model

We consider a federated learning framework as shown in Fig. 1 that consists of one central server and $N$ data-owning clients. Each client $i \in [N]$ holds a local dataset $(\mathbf{X}_i, \mathbf{Y}_i)$ of size $m_i$, where $\mathbf{X}_i \in \mathbb{R}^{m_i \times d_x}$ represents the set of input features of dimension $d_x$ and $\mathbf{Y}_i \in \mathbb{R}^{m_i \times d_y}$ corresponds to the output vector of dimension $d_y$. Accordingly, the size of the global dataset $(\mathbf{X}, \mathbf{Y})$ which concatenates all local datasets $(\mathbf{X}_i, \mathbf{Y}_i), \forall i \in [N]$ is denoted as $m \triangleq \sum_{i=1}^{N} m_i$. The clients aim to jointly train a neural network based on their local datasets without sharing private data samples. Particularly, the gradients are computed locally and aggregated globally. However, the local data may be highly heterogeneous, and the clients may drop out at any time unexpectedly, which makes the training process unstable. Our goal is to improve the convergence performance by secret data sharing while preserving the privacy of local datasets.

## 3.1 Lagrange Coded Computing for Federated Learning

The Lagrange coded computing (LCC) framework enables private computing in distributed settings to provide resiliency and efficiency [22]. The key idea is using Lagrange coding to encode the data for redundant distributed computing, which fits nicely with federated learning due to its dropout-resiliency and privacy requirement. Specifically, the clients share their encoded datasets with each other and perform gradient computation over the encoded samples. The server decodes the aggregate gradient after receiving the uploaded computation results from clients. To provide a strong privacy guarantee for the datasets and correctly decode the gradient at the server, the gradient function should be a polynomial function in a finite field. However, existing neural networks cannot satisfy this requirement, since the datasets are in the real field and the gradients are non-polynomial.

**Polynomial integer neural networks.** We define a class of polynomial integer neural networks (PINNs) to ensure that the gradient is a polynomial function in a finite field $\mathbb{F}_p$ with a prime number $p$. First, we transform the dataset $(\mathbf{X}, \mathbf{Y})$ from the real domain to the finite domain $(\overline{\mathbf{X}}, \overline{\mathbf{Y}})$. Besides, a PINN consists of affine transformation layers (e.g., fully connected layers and convolutional layers) and utilizes the quadratic function as the activation function. The model parameters of PINNs are defined in the integer set $\mathbb{Z}_p \triangleq \{-\lfloor \frac{p+1}{2} \rfloor, \ldots, \lfloor \frac{p-1}{2} \rfloor\}$. Given a feed-forward function $\boldsymbol{f}(\overline{\mathbf{X}}; \mathbf{w})$ and selecting the mean squared error (MSE) as the loss function, the gradient of the input samples is a multivariate polynomial with integer coefficients, i.e., $\boldsymbol{g}(\overline{\mathbf{X}}, \overline{\mathbf{Y}}; \mathbf{w}) \triangleq \nabla_{\mathbf{w}} \|\overline{\mathbf{Y}} - \boldsymbol{f}(\overline{\mathbf{X}}; \mathbf{w})\|_2^2 \in \mathbb{Z}^{d_w}$, where $d_w$ represents the number of model parameters. Denoting the number of quadratic activation layers as $L$, the degree of the gradient function[3] $\boldsymbol{g}(\overline{\mathbf{X}}, \overline{\mathbf{Y}}; \mathbf{w})$ is $\deg(\boldsymbol{g}) = 2^{L+1}$.

In particular, to avoid wrap-around when computing gradient in the finite field $\mathbb{F}_p$, we assume the prime number $p$ is sufficiently large without leading to overflow errors in the integer set $\mathbb{Z}_p$.

**Lagrange coding.** The proposed DReS-FL framework uses Lagrange polynomials to achieve a $D$-resilient, $T$-private, and $K$-efficient coding scheme. $D$-Resiliency means that the global gradient can be decoded by the server in the presence of up to $D$ client dropouts. $T$-privacy denotes that no information about local datasets can be inferred from the encoded data even if up to $T$ clients collude. $K$-efficiency corresponds to the complexity of the coding scheme. Specifically, each private dataset is split into $K$ shards in Lagrange coding, and the size of the encoded dataset is proportional to $1/K$. Therefore, increasing the value of $K$ reduces the communication overhead of data sharing. The following theorem characterizes the $(D, T, K)$-achievable coding scheme, and its proof is available in Section IV of [22].

**Theorem 1.** *Given the client number $N$ and the degree of the gradient function $\deg(\boldsymbol{g})$, a $D$-resilient, $T$-private, and $K$-efficient Lagrange coding scheme is achievable, as long as*

$$D + \deg(\boldsymbol{g})(K + T - 1) + 1 \leq N. \tag{1}$$

**Remark 1.** As shown in Theorem 1, there is a tradeoff among resiliency ($D$), privacy ($T$), and efficiency ($K$). As the sum of $T$ and $K$ increases, the proposed framework tolerates fewer client dropouts. Specifically, the maximum value of $D$ is $N - 1 - \deg(\boldsymbol{g})$ by setting $T = K = 1$.

**Remark 2.** Setting the privacy parameter $T \geq 1$, the gradient computation over the encoded samples leaks no private information according to the data process inequality. This implies that the proposed DReS-FL framework achieves the same privacy guarantee as the secure aggregation protocols. Specifically, the server learns no information about the private dataset from a single client's computation result.

## 4 DReS-FL Framework

DReS-FL consists of two main phases, as shown in Fig. 1. In the first phase, the private datasets are transformed from the real domain to the finite field, and data-owning clients secretly share datasets by Lagrange coding. Then, the server and the clients train a PINN iteratively via (1) local gradient computations and (2) gradient decoding and model updating.

---

[3]More details about how to calculate the degree of the gradient are deferred to Appendix E.

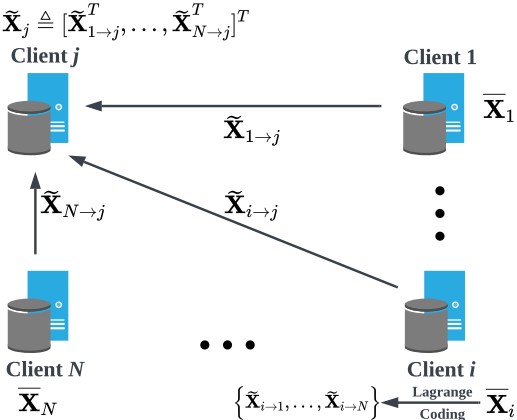

$$\widetilde{\mathbf{X}}_j \triangleq [\widetilde{\mathbf{X}}_{1\to j}^T, \dots, \widetilde{\mathbf{X}}_{N\to j}^T]^T$$

**Client $j$**

**Client 1**

$\overline{\mathbf{X}}_1$

$\widetilde{\mathbf{X}}_{1\to j}$

$\widetilde{\mathbf{X}}_{N\to j}$

$\widetilde{\mathbf{X}}_{i\to j}$

**Client $N$**

$\overline{\mathbf{X}}_N$

**Client $i$**

$\left\{\widetilde{\mathbf{X}}_{i\to 1}, \dots, \widetilde{\mathbf{X}}_{i\to N}\right\} \xleftarrow{\text{Lagrange Coding}} \overline{\mathbf{X}}_i$

Figure 2: The secret data sharing scheme in the DReS-FL framework. Every client $i \in [N]$ secretly shares the local dataset $\overline{\mathbf{X}}_i$ to other clients $j \in [N]\backslash\{i\}$ by sending the encoded samples $\widetilde{\mathbf{X}}_{i\to j}$. The client $j$ will receive $\widetilde{\mathbf{X}}_j \triangleq [\widetilde{\mathbf{X}}_{1\to j}^T, \dots, \widetilde{\mathbf{X}}_{N\to j}^T]^T$ as an encoded version of the global dataset.

## 4.1 Data Transformation and Secret Sharing

To guarantee information-theoretic privacy, each client has to mask the datasets in a finite field $\mathbb{F}_p$ using uniformly random matrices. Firstly, the local datasets $(\mathbf{X}_i, \mathbf{Y}_i)$ are converted from the real domain to the finite field $(\overline{\mathbf{X}}_i, \overline{\mathbf{Y}}_i)$. Considering an element-wise function $\phi(z) = z + c$ that transforms a real value to a non-negative number by adding a proper scalar $c$[4], we define $\overline{\mathbf{X}} \triangleq Round(2^l \cdot \phi(\mathbf{X}))$, where the rounding operation is element-wise that quantizes each entry to its closest integer, and $l \in \mathbb{Z}$ controls the quantization loss. We adopt the notation $\overline{\mathcal{D}}$ to represent the *global dataset*, which is the concatenation of all the local datasets $(\overline{\mathbf{X}}_i, \overline{\mathbf{Y}}_i)$ for $i \in [N]$.

After converting the private datasets to the finite field, the clients adopt a $(D, T, K)$-achievable Lagrange coding to encode local data for secret sharing. First, each client $i \in [N]$ partitions its local dataset to $K$ shards as $\overline{\mathbf{X}}_i \triangleq [\overline{\mathbf{X}}_i^{(1)T}, \dots, \overline{\mathbf{X}}_i^{(K)T}]^T$ and $\overline{\mathbf{Y}}_i \triangleq [\overline{\mathbf{Y}}_i^{(1)T}, \dots, \overline{\mathbf{Y}}_i^{(K)T}]^T$. Assuming that $m_i$ is divisible by $K$, we have $\overline{\mathbf{X}}_i^{(k)} \in \mathbb{F}_p^{\frac{m_i}{K} \times d_x}$ and $\overline{\mathbf{Y}}_i^{(k)} \in \mathbb{F}_p^{\frac{m_i}{K} \times d_y}$ for $k \in [K]$. A large value of $K$ helps to reduce the complexity in secret data sharing. Then, the clients add padding from $T$ uniform random masks to the data samples for privacy protection. Each client $i \in [N]$ forms the following polynomials $\mathbf{u}_i : \mathbb{F}_p \to \mathbb{F}_p^{\frac{m_i}{K} \times d_x}$ and $\mathbf{v}_i : \mathbb{F}_p \to \mathbb{F}_p^{\frac{m_i}{K} \times d_y}$ of degree $K + T - 1$ to encode the local dataset:

$$\mathbf{u}_i(z) \triangleq \sum_{k\in[K]} \overline{\mathbf{X}}_i^{(k)} \cdot \prod_{j\in[K+T]\backslash\{k\}} \frac{z - \beta_j}{\beta_k - \beta_j} + \sum_{k=K+1}^{K+T} \mathbf{U}_i^{(k)} \cdot \prod_{j\in[K+T]\backslash\{k\}} \frac{z - \beta_j}{\beta_k - \beta_j}, \qquad (2)$$

$$\mathbf{v}_i(z) \triangleq \sum_{k\in[K]} \overline{\mathbf{Y}}_i^{(K)} \cdot \prod_{j\in[K+T]\backslash\{k\}} \frac{z - \beta_j}{\beta_k - \beta_j} + \sum_{k=K+1}^{K+T} \mathbf{V}_i^{(k)} \cdot \prod_{j\in[K+T]\backslash\{k\}} \frac{z - \beta_j}{\beta_k - \beta_j}, \qquad (3)$$

where $\{\mathbf{U}_i^{(k)}\}$'s and $\{\mathbf{V}_i^{(k)}\}$'s are random noise matrices uniformly sampled from $\mathbb{F}_p^{\frac{m}{K} \times d_x}$ and $\mathbb{F}_p^{\frac{m}{K} \times d_y}$, respectively. These matrices mask the local datasets and provide a privacy guarantee against up to $T$ colluding workers. The clients and the server agree on $K + T$ distinct elements $\{\beta_1, \dots, \beta_{K+T}\}$ from the finite field $\mathbb{F}_p$ in advance. Particularly, setting $z = \beta_k$ for $k \in [K]$, we reconstruct the data shard $(\mathbf{u}_i(\beta_k), \mathbf{v}_i(\beta_k)) = (\overline{\mathbf{X}}_i^{(k)}, \overline{\mathbf{Y}}_i^{(k)})$. All the clients use the same $N$ distinct elements $\{\alpha_1, \dots, \alpha_N\}$ selected from $\mathbb{F}_p$ to encode the private datasets, where $\{\alpha_j\}_{j\in[N]} \cap \{\beta_k\}_{k\in[K+T]} = \varnothing$. Each client $i$ obtains $N$ encoded datasets $(\widetilde{\mathbf{X}}_{i\to j}, \widetilde{\mathbf{Y}}_{i\to j}) \triangleq (\mathbf{u}_i(\alpha_j), \mathbf{v}_i(\alpha_j))$ for $j \in [N]$, where each $(\widetilde{\mathbf{X}}_{i\to j}, \widetilde{\mathbf{Y}}_{i\to j})$ is sent to client $j$ from client $i$. All the received encoded

---

[4]The scalar $c$ could be the absolute value of the minimum entry in dataset, which is set to 0 in the experiments.

Table 1: Primary notations and descriptions

| Notation | Description | Notation | Description |
|---|---|---|---|
| $(\overline{\mathbf{X}}_i, \overline{\mathbf{Y}}_i)$ | Transformed dataset at client $i$ in a finite filed $\mathbb{F}_p$ | $\overline{\mathcal{D}}$ | The global dataset that is a concatenation of clients' datasets $(\overline{\mathbf{X}}_i, \overline{\mathbf{Y}}_i)$ for $i \in [N]$ |
| $(\overline{\mathbf{X}}_i^{(k)}, \overline{\mathbf{Y}}_i^{(k)})$ | $k$-th data shard at client $i$ | $(\widetilde{\mathbf{X}}_{i \to j}, \widetilde{\mathbf{Y}}_{i \to j})$ | Encoded dataset sent from client $i$ to client $j$ |
| $(\widetilde{\mathbf{X}}_j, \widetilde{\mathbf{Y}}_j)$ | Concatenation of received encoded datasets $(\widetilde{\mathbf{X}}_{i \to j}, \widetilde{\mathbf{Y}}_{i \to j})$ for $i \in [N]$ at client $j$ | $\mathbf{C}^{(t)}, \mathcal{I}_t$ | Row selection matrix $\mathbf{C}^{(t)}$ for data sampling in round $t$ and the corresponding index set $\mathcal{I}_t$ |
| $(\widetilde{\mathbf{X}}_j^{(\mathcal{I}_t)}, \widetilde{\mathbf{Y}}_j^{(\mathcal{I}_t)})$ | Local mini-batch sampled from $(\widetilde{\mathbf{X}}_j, \widetilde{\mathbf{Y}}_j)$ at client $j$ in round $t$ based on $\mathcal{I}_t$ | $(\overline{\mathbf{X}}_{\mathcal{I}_t}^{(k)}, \overline{\mathbf{Y}}_{\mathcal{I}_t}^{(k)})$ | $k$-th global mini-batch sampled from $\overline{\mathcal{D}}$ in round $t$ based on $\mathcal{I}_t$ |

---

**Algorithm 1** DReS-FL

---

**Input:** Local datasets $(\mathbf{X}_i, \mathbf{Y}_i)$ for $i \in [N]$, batch size $b$, initialized parameters $\mathbf{w}^{(0)} \in \mathbb{Z}_p^{d_w}$, distinct elements $\{\alpha_j\}_{j \in [N]}$ and $\{\beta_k\}_{k \in [K+T]}$, prime number $p$, training round $\tau$, learning rate $\eta$.
**Output:** Model parameter $\mathbf{w}^{(\tau)}$.
1: Clients encode the local datasets according to (2) and (3) and deliver them to other clients.
2: **for** $t = 1, 2, \ldots, \tau$ **do**
3:     Server sends the model parameters $\mathbf{w}^{(t)}$ to the clients.
4:     **for** $j = 1, \ldots, N$ **do**
5:         Client $j$ performs gradient computation on mini-batches $(\widetilde{\mathbf{X}}_j^{(\mathcal{I}_t)}, \widetilde{\mathbf{Y}}_j^{(\mathcal{I}_t)})$.
6:         Upload local computation results $\widetilde{g}(\widetilde{\mathbf{X}}_j^{(\mathcal{I}_t)}, \widetilde{\mathbf{Y}}_j^{(\mathcal{I}_t)}; \mathbf{w}^{(t)})$ to the server.
7:     **end for**
8:     **if** Server receives at least $\deg(g)(K + T - 1) + 1$ uploads **then**
9:         Decode $K$ global gradients $\widetilde{g}(\overline{\mathbf{X}}_{\mathcal{I}_t}^{(k)}, \overline{\mathbf{Y}}_{\mathcal{I}_t}^{(k)}; \mathbf{w}^{(t)})$ for $k \in [K]$ by polynomial interpolation.
10:         Convert gradients from the finite field to the integral domain $g(\overline{\mathbf{X}}_{\mathcal{I}_t}^{(k)}, \overline{\mathbf{Y}}_{\mathcal{I}_t}^{(k)}; \mathbf{w}^{(t)})$ by (6).
11:         Update the model by (7) based on the aggregate gradient $\sum_{k=1}^K g(\overline{\mathbf{X}}_{\mathcal{I}_t}^{(k)}, \overline{\mathbf{Y}}_{\mathcal{I}_t}^{(k)}; \mathbf{w}^{(t)})$.
12:     **end if**
13: **end for**

---

datasets at client $j$ are represented as $(\widetilde{\mathbf{X}}_j, \widetilde{\mathbf{Y}}_j)$, where $\widetilde{\mathbf{X}}_j \triangleq [\widetilde{\mathbf{X}}_{1 \to j}^T, \ldots, \widetilde{\mathbf{X}}_{N \to j}^T]^T \in \mathbb{F}_p^{\widetilde{m} \times d_x}$ and $\widetilde{\mathbf{Y}}_j \triangleq [\widetilde{\mathbf{Y}}_{1 \to j}^T, \ldots, \widetilde{\mathbf{Y}}_{N \to j}^T]^T \in \mathbb{F}_p^{\widetilde{m} \times d_y}$ for $j \in [N]$. Accordingly, the number of samples in the encoded dataset is $\widetilde{m} \triangleq \frac{1}{K} \sum_{i=1}^n m_i$. Fig. 2 demonstrates the secret data sharing scheme.

## 4.2 Federated Training

**Local Gradient Computation.** The server randomly initializes a PINN at the beginning of the training process, and the model parameters are constrained to an integer set $\mathbb{Z}_p$ during the training process. In each communication round, the server sends the model parameters to the clients, and they compute the stochastic gradient over the mini-batches with size $b$. Particularly, we assume that all the clients use the same row selection matrix $\mathbf{C}^{(t)} \in \{0, 1\}^{b \times \widetilde{m}}$ for data sampling in each round $t$[5], and the mini-batch at each client $j \in [N]$ is determined by $[\widetilde{\mathbf{X}}_j^{(\mathcal{I}_t)}, \widetilde{\mathbf{Y}}_j^{(\mathcal{I}_t)}] \triangleq \mathbf{C}^{(t)}[\widetilde{\mathbf{X}}_j, \widetilde{\mathbf{Y}}_j]$. Here, $\mathcal{I}_t = \{l_1^{(t)}, \ldots, l_b^{(t)}\} \subseteq [\widetilde{m}]$ is a randomly selected index set in the $t$-th round with $l_i \in [\widetilde{m}]$ for $i \in [b]$. The entries of $\mathbf{C}^{(t)}$ satisfy $\mathbf{C}_{i,l_i}^{(t)} = 1$ for $i \in [b]$, and other entries are set to zero. Each client $j$ computes the stochastic gradient $\widetilde{g}(\widetilde{\mathbf{X}}_j^{(\mathcal{I}_t)}, \widetilde{\mathbf{Y}}_j^{(\mathcal{I}_t)}; \mathbf{w}^{(t)}) \triangleq g(\widetilde{\mathbf{X}}_j^{(\mathcal{I}_t)}, \widetilde{\mathbf{Y}}_j^{(\mathcal{I}_t)}; \mathbf{w}^{(t)}) \mod p$ in the finite field, and uploads the result to the server. Particularly, each $\widetilde{g}(\widetilde{\mathbf{X}}_j^{(\mathcal{I}_t)}, \widetilde{\mathbf{Y}}_j^{(\mathcal{I}_t)}; \mathbf{w}^{(t)})$ amounts to an evaluation of the polynomial $\widetilde{g}(\mathbf{u}_{\mathcal{I}_t}(z), \mathbf{v}_{\mathcal{I}_t}(z); \mathbf{w}^{(t)})$ at the point $z = \alpha_j$, where two $(K + T - 1)$-degree polynomial functions $\mathbf{u}_{\mathcal{I}_t} : \mathbb{F}_p \to \mathbb{F}_p^{b \times d_x}$ and $\mathbf{v}_{\mathcal{I}_t} : \mathbb{F}_p \to \mathbb{F}_p^{b \times d_y}$ are defined as

---

[5]This can be achieved by setting the same random seed across all the clients. The weighted sampling method has been adopted in this work, where the number of sampled data from $\widetilde{\mathbf{X}}_{i \to j}$ is proportional to $m_i$ for $i \in [N]$. Note that other sampling schemes can also be applied in DReS-FL.

follows:

$$\mathbf{u}_{\mathcal{I}_t}(z) \triangleq \sum_{k \in [K]} \overline{\mathbf{X}}_{\mathcal{I}_t}^{(k)} \cdot \prod_{j \in [K+T] \setminus \{k\}} \frac{z - \beta_j}{\beta_k - \beta_j} + \sum_{k=K+1}^{K+T} \mathbf{U}_{\mathcal{I}_t}^{(k)} \cdot \prod_{j \in [K+T] \setminus \{k\}} \frac{z - \beta_j}{\beta_k - \beta_j}, \quad (4)$$

$$\mathbf{v}_{\mathcal{I}_t}(z) \triangleq \sum_{k \in [K]} \overline{\mathbf{Y}}_{\mathcal{I}_t}^{(k)} \cdot \prod_{j \in [K+T] \setminus \{k\}} \frac{z - \beta_j}{\beta_k - \beta_j} + \sum_{k=K+1}^{K+T} \mathbf{V}_{\mathcal{I}_t}^{(k)} \cdot \prod_{j \in [K+T] \setminus \{k\}} \frac{z - \beta_j}{\beta_k - \beta_j}, \quad (5)$$

with

$$\overline{\mathbf{X}}_{\mathcal{I}_t}^{(k)} = \mathbf{C}^{(t)} [\overline{\mathbf{X}}_1^{(k)T}, \dots, \overline{\mathbf{X}}_N^{(k)T}]^T \in \mathbb{F}_p^{b \times d_x}, \quad \mathbf{U}_{\mathcal{I}_t}^{(k)} = \mathbf{C}^{(t)} [\mathbf{U}_1^{(k)T}, \dots, \mathbf{U}_N^{(k)T}]^T \in \mathbb{F}_p^{b \times d_x},$$

$$\overline{\mathbf{Y}}_{\mathcal{I}_t}^{(k)} = \mathbf{C}^{(t)} [\overline{\mathbf{Y}}_1^{(k)T}, \dots, \overline{\mathbf{Y}}_N^{(k)T}]^T \in \mathbb{F}_p^{b \times d_y}, \quad \mathbf{V}_{\mathcal{I}_t}^{(k)} = \mathbf{C}^{(t)} [\mathbf{V}_1^{(k)T}, \dots, \mathbf{V}_N^{(k)T}]^T \in \mathbb{F}_p^{b \times d_y}.$$

Every $(\overline{\mathbf{X}}_{\mathcal{I}_t}^{(k)}, \overline{\mathbf{Y}}_{\mathcal{I}_t}^{(k)}) = (\mathbf{u}_{\mathcal{I}_t}(\beta_k), \mathbf{v}_{\mathcal{I}_t}(\beta_k))$ for $k \in [K]$ is a *global mini-batch* selected from the global dataset $\overline{\mathcal{D}}$.

**Gradient Decoding and Model Updating.** According to (4) and (5), the server can obtain the global gradients $\widetilde{\boldsymbol{g}}(\overline{\mathbf{X}}_{\mathcal{I}_t}^{(k)}, \overline{\mathbf{Y}}_{\mathcal{I}_t}^{(k)}; \mathbf{w}^{(t)})$ for $k \in [K]$ by evaluating the polynomial $\widetilde{\boldsymbol{g}}(\mathbf{u}_{\mathcal{I}_t}(z), \mathbf{v}_{\mathcal{I}_t}(z); \mathbf{w}^{(t)})$ at the point $z = \beta_k$. But, the server needs to first recover the coefficients of this polynomial, which is a composition of the encoding polynomials $(\mathbf{u}_{\mathcal{I}_t}(z), \mathbf{v}_{\mathcal{I}_t}(z))$ and the gradient function $\widetilde{\boldsymbol{g}}$. As the degree of the composite polynomial is $\deg(\boldsymbol{g})(K+T-1)$, the server requires at least $\deg(\boldsymbol{g})(K+T-1)+1$ local computation results (i.e., evaluation points) to interpolate it[6]. This implies that the proposed DReS-FL framework can tolerate at most $D = N - \deg(\boldsymbol{g})(K+T-1) - 1$ client dropouts.

After decoding the global gradients, the server converts them from the finite field to the integer set $\mathbb{Z}_p$ by $\boldsymbol{g}(\overline{\mathbf{X}}_{\mathcal{I}_t}^{(k)}, \overline{\mathbf{Y}}_{\mathcal{I}_t}^{(k)}; \mathbf{w}^{(t)}) = \psi(\widetilde{\boldsymbol{g}}(\overline{\mathbf{X}}_{\mathcal{I}_t}^{(k)}, \overline{\mathbf{Y}}_{\mathcal{I}_t}^{(k)}; \mathbf{w}^{(t)}))$, where $\psi(z)$ is an element-wise function defined as follows:

$$\psi(z) = \begin{cases} z & \text{if} \quad 0 \le z < \frac{p-1}{2}, \\ z - p & \text{if} \quad \frac{p-1}{2} \le z < p. \end{cases} \quad (6)$$

As we assume that the prime number $p$ is sufficiently large, the converted gradients do not have overflow errors. Thus, the central sever updates the global model by $\mathbf{w}^{(t+1)} = \mathbf{w}^{(t)} - Q(\frac{\eta}{bK} \sum_{k=1}^{K} \boldsymbol{g}(\overline{\mathbf{X}}_{\mathcal{I}_t}^{(k)}, \overline{\mathbf{Y}}_{\mathcal{I}_t}^{(k)}; \mathbf{w}^{(t)}))$, where $\eta$ denotes the learning rate and $bK$ represents the *global batch size*[7]. $Q(z)$ is a stochastic quantization function to ensure the model parameters are in the integer set $\mathbb{Z}_p$ after updating, which is defined as follows:

$$Q(z) = \begin{cases} \lfloor z \rfloor & \text{with probability } 1 - (z - \lfloor z \rfloor) \\ \lfloor z \rfloor + 1 & \text{with probability } z - \lfloor z \rfloor. \end{cases} \quad (7)$$

Besides, the probability of rounding $z$ to $\lfloor z \rfloor$ is proportional to the proximity of $z$ to $\lfloor z \rfloor$ so that the stochastic rounding is unbiased. The overall procedure is summarized in Algorithm 1.

## 5   Convergence Analysis

In this section we characterize the convergence performance of PINNs, which relies on the fact that the global gradients in the training process are unbiased. Define the empirical risk as $\ell(\mathbf{w}) \triangleq \mathbb{E}_{(\overline{\mathbf{X}}, \overline{\mathbf{Y}}) \sim \overline{\mathcal{D}}} \|\overline{\mathbf{Y}} - \boldsymbol{f}(\overline{\mathbf{X}}; \mathbf{w})\|_2^2$ and the corresponding gradient as $\boldsymbol{g}_e(\mathbf{w}) \triangleq \mathbb{E}_{(\overline{\mathbf{X}}, \overline{\mathbf{Y}}) \sim \overline{\mathcal{D}}}[\boldsymbol{g}(\overline{\mathbf{X}}, \overline{\mathbf{Y}}; \mathbf{w})]$. The variables $(\overline{\mathbf{X}}, \overline{\mathbf{Y}})$ are drawn from the distribution of the global dataset $\overline{\mathcal{D}}$. To prove that DReS-FL guarantees convergence to the optimal model parameters, we first present the following assumptions to facilitate the analysis.

**Assumption 1.** *(L-smoothness) There exists a constant $L > 0$ such that for all $\mathbf{w}_1, \mathbf{w}_2 \in \mathbb{Z}_p^{d_w}$, we have $\|\boldsymbol{g}_e(\mathbf{w}_1) - \boldsymbol{g}_e(\mathbf{w}_2)\|_2 \le L \|\mathbf{w}_1 - \mathbf{w}_2\|_2$.*

---

[6]Note that if the server cannot receive enough results due to the client dropouts, the training protocol continues to the next epoch without gradient decoding and model updating.

[7]The global batch size corresponds to the number of samples used for gradient computations in each round.

Table 2: Test accuracy $(\%)$ of different methods. Each experiment is repeated five times. Best results are shown in italic and second best results are in bold.

| Dataset | MNIST | Fashion-MNIST | EMNIST | CIFAR-10 | CIFAR-100 | SVHN |
|---|---|---|---|---|---|---|
| FedAvg | $96.17 \pm 0.05$ | $81.20 \pm 0.07$ | $71.50 \pm 0.28$ | $89.54 \pm 0.09$ | $67.71 \pm 0.26$ | $83.82 \pm 0.20$ |
| FedAvg-IS | $97.06 \pm 0.10$ | $85.94 \pm 0.16$ | $77.09 \pm 0.34$ | $89.83 \pm 0.07$ | $68.92 \pm 0.14$ | $85.27 \pm 0.09$ |
| SCAFFOLD | $71.89 \pm 3.92$ | $55.22 \pm 1.83$ | $55.15 \pm 5.95$ | $54.17 \pm 9.13$ | $29.97 \pm 1.73$ | $51.27 \pm 3.43$ |
| DReS-FL (Ours) | $\mathbf{97.38 \pm 0.08}$ | $\mathbf{86.60 \pm 0.32}$ | $\mathbf{78.04 \pm 0.29}$ | $\mathbf{90.31 \pm 0.19}$ | $\mathbf{69.15 \pm 0.27}$ | $\mathbf{86.04 \pm 0.15}$ |
| Centralized | $\mathit{97.99 \pm 0.04}$ | $\mathit{89.02 \pm 0.11}$ | $\mathit{82.45 \pm 0.23}$ | $\mathit{90.37 \pm 0.12}$ | $\mathit{71.12 \pm 0.09}$ | $\mathit{86.18 \pm 0.03}$ |

**Assumption 2.** *(Unbiased and variance-bounded stochastic gradient) There exists a constant $\sigma > 0$ such that any stochastic gradient $g(\overline{\mathbf{X}}_{\mathcal{I}_t}^{(k)}, \overline{\mathbf{Y}}_{\mathcal{I}_t}^{(k)}; \mathbf{w}^{(t)})$ satisfies $\mathbb{E}\left[\frac{1}{b}g(\overline{\mathbf{X}}_{\mathcal{I}_t}^{(k)}, \overline{\mathbf{Y}}_{\mathcal{I}_t}^{(k)}; \mathbf{w}^{(t)})\right] = \boldsymbol{g}_e(\mathbf{w}^{(t)})$ and $\mathbb{E}\left[\|\frac{1}{b}g(\overline{\mathbf{X}}_{\mathcal{I}_t}^{(k)}, \overline{\mathbf{Y}}_{\mathcal{I}_t}^{(k)}; \mathbf{w}^{(t)}) - \boldsymbol{g}_e(\mathbf{w}^{(t)})\|^2\right] \leq \sigma^2.$*

**Assumption 3.** *(Unbiased and variance-bounded rounding operation) There exists a constant $\gamma > 0$ such that for any $z \in \mathbb{R}$, the stochastic quantization operation $Q(\cdot)$ satisfies $\mathbb{E}\left[Q(z)\right] = z$ and $\mathbb{E}\left[\|Q(z) - z\|^2\right] \leq \gamma^2 z^2.$*

With the above preparations, we have the following theorem which ensures the convergence. The proof is deferred to Appendix A.

**Theorem 2.** *(Convergence) Denote $\mathbf{w}^*$ as the first-order optimal solution. With Assumption 1-3, selecting the learning rate as $\eta = \mathcal{O}\left(1/\sqrt{\tau'}\right)$ such that $\Psi \triangleq 1 - \eta L/2 - \eta\gamma^2 L/2 > 0$, after $\tau'$ times of model updates, we have:*

$$\frac{1}{\tau'}\sum_{t=1}^{\tau'}\mathbb{E}\left[\|\boldsymbol{g}_e(\mathbf{w}^{(t)})\|^2\right] \leq \frac{\ell(\mathbf{w}^{(0)}) - \ell(\mathbf{w}^*)}{\eta\tau'\Psi} + \frac{\eta^2 L\sigma^2}{2bK\Psi}(\gamma^2 + 1). \qquad (8)$$

# 6 Experiments

## 6.1 Experimental Setup

**Dataset.** We evaluate our proposed algorithm on several benchmark datasets: MNIST [45], Fashion-MNIST [46], EMNIST (Balanced) [47], CIFAR-10 [48], CIFAR-100 [48], and SVHN [49]. Specifically, the extra training samples in the SVHN dataset are not utilized. To simulate the non-IID data distribution, we assume there are $N = 20$ clients in the learning system and adopt the skewed label partition [50] to shuffle the datasets. Specifically, we sort a dataset by the labels, divide it into $N$ shards, and assign one shard to each client. To simulate the client dropouts in the training process, we consider an extreme scenario, where the dropout rate of each client is set to 0.99 with a probability of 0.5 or is uniformly sampled from $[0, 0.1]$ otherwise. More details of the datasets are deferred to Appendix B.

**Model structures.** We adopt a multi-layer perception (MLP) with two hidden layers for the image classification tasks on MNIST, Fashion-MNIST, and EMNIST datasets. Each hidden layer contains 64 neurons. For CIFAR-10, CIFAR-100, and SVHN datasets, we resize the input images from $32 \times 32$ to $224 \times 224$ and adopt the convolutional layers of a pretrained VGG model to extract 25088-dimensional features. To classify the extracted features, we select a two-layer MLP model with 4096 hidden units each. The baseline methods train the neural networks on the real field and select the rectified linear unit (ReLU) function as the activation function. In each communication round, clients perform one SGD step for the local model update.

**DReS-FL.** Our method adopts the same size PINNs to replace MLPs in the federated training, and the degree of gradient is $\deg(\boldsymbol{g}) = 8$. Particularly, the extracted features from the last convolutional layer of VGG19 are secretly shared with other clients. We set the parameters $K = 1$ and $T = 1$ in the Lagrange coding, and the minimum number of clients needed to decode the global gradient is 9.

**Baselines.** In the experiments, data-centric approaches [11, 15, 17, 33, 34, 35] are not compared since some of them [11, 15, 17] lack strong privacy guarantees while others [33, 34, 35] cannot support federated neural network training with multiple clients. We select algorithm-based methods as baselines, including FedAvg [1], FedAvg with importance sampling (FedAvg-IS) [51, 52], and

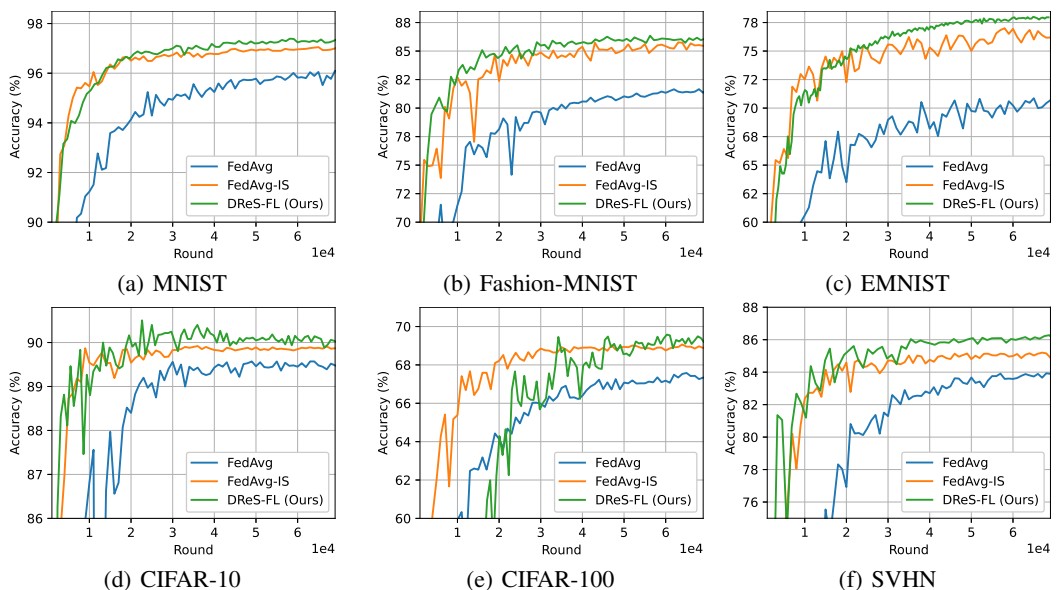

| (a) MNIST | (b) Fashion-MNIST | (c) EMNIST |
| (d) CIFAR-10 | (e) CIFAR-100 | (f) SVHN |

Figure 3: Test accuracies on different datasets.

SCAFFOLD [2], since these methods can be easily combined with secure aggregation methods[8]. Particularly, we assume that the FedAvg-IS method knows the dropout distribution, and the local computation results are weighted by the participation probability (i.e., 1 - dropout probability) to mitigate bias in aggregation. Besides, we also select the centralized training scheme as a performance upper bound, where the server can access all the clients' datasets for model training.

## 6.2 Performance Evaluation

The experimental results are shown in Table 2 and Fig. 3. The FedAvg method achieves worse performance than the centralized training scheme. This is attributed to the non-IID data and client dropouts. The FedAvg-IS method improves the test accuracy compared with FedAvg, but there is still a noticeable performance gap with the centralized training scheme. It shows that using the knowledge of dropout distribution can partially compensate for the biases in the aggregated models, but the local data distributions are still heterogeneous and degrade the performance. Besides, SCAFFOLD has a low accuracy on all the settings. As the frequency of updating local control variates is low, the estimation of the update direction is highly inaccurate such that the model does not converge as shown in Fig. 4. These results are consistent with the findings in [10]. Our

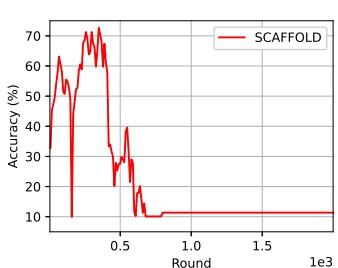

Figure 4: Test accuracy of SCAFFOLD on MNIST.

DReS-FL method is superior to all the baseline methods as the server can obtain global gradients after polynomial interpolation. In addition, DReS-FL achieves comparable performance to the centralized training scheme on some datasets, which demonstrates the effectiveness of our proposed framework in solving the non-IID and dropout problems.

## 7 Conclusions

This paper proposed a Dropout-Resilient Secure Federated Learning (DReS-FL) framework via Lagrange coded computing (LCC) to simultaneously solve the data heterogeneity and dropout problems of FL, while providing privacy guarantees for the local datasets. The polynomial integer neural networks (PINNs) have been constructed to ensure that the server can correctly decode the

---

[8]Note that the secure aggregation mechanism has not been applied in the experiments, since the quantization step in secure aggregation may degrade the performance of the baselines.

global gradient without privacy leakage. Extensive experimental results validated the effectiveness of the proposed method. Potential limitations of our method include that the degree of the gradient in a PINN increases exponentially with the number of layers, which hinders training a deep model for complex tasks. Besides, performing multiple local SGD steps largely increases the finite field size as the range of results grows exponentially with the number of multiplications, and thus it will lead to substantial communication overhead in model transmission. Despite some limitations, we believe DReS-FL is a promising framework for many practical FL application scenarios given its effectiveness in resolving both the non-IID and client dropout problems, while with strong privacy guarantees.

## Acknowledgement

The work of Songze Li is in part supported by the National Nature Science Foundation of China (NSFC) Grant 62106057, and Foshan HKUST Projects FSUST20-FYTRI04B.

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
