# Appendix

## A   Proof of Theorem 2

For simplicity, we denote $\boldsymbol{g}^{(t)} = \frac{1}{bK} \sum_{k=1}^{K} \boldsymbol{g}(\overline{\mathbf{X}}_{\mathcal{I}_t}^{(k)}, \overline{\mathbf{Y}}_{\mathcal{I}_t}^{(k)}; \mathbf{w}^{(t)})$ in this section. The server updates the global model by $\mathbf{w}^{(t+1)} = \mathbf{w}^{(t)} - Q(\eta \boldsymbol{g}^{(t)})$ in each round $t$ after receiving $\deg(\boldsymbol{g})(K+T-1)+1$ uploads from clients. We first provide an important lemma to show that the model update $Q(\eta \boldsymbol{g}^{(t)})$ on the server is an unbiased estimate of $\eta \boldsymbol{g}_e(\mathbf{w}^{(t)})$.

**Lemma 1.** *(Unbiased and variance-bounded model update) In the t-th round, the model update* $Q(\eta \boldsymbol{g}^{(t)})$ *has the following properties:*

$$\mathbb{E}\left[Q(\eta \boldsymbol{g}^{(t)})\right] = \eta \boldsymbol{g}_e(\mathbf{w}), \tag{9}$$

$$\mathbb{E}\left[\|Q(\eta \boldsymbol{g}^{(t)}) - \eta \boldsymbol{g}_e(\mathbf{w})\|^2\right] \leq (\gamma^2 + 1)\eta^2 \frac{\sigma^2}{bK} + \gamma^2 \eta^2 \left\|\boldsymbol{g}_e(\mathbf{w}^{(t)})\right\|^2. \tag{10}$$

*Proof.* According to Assumption 2-3, we directly obtain that $\mathbb{E}\left[Q(\eta \boldsymbol{g}^{(t)})\right] = \mathbb{E}\left[\eta \boldsymbol{g}^{(t)}\right] = \boldsymbol{g}_e(\mathbf{w})$. Since the batch sampling and rounding operation cause independent errors, the variance is upper bounded as follows:

$$
\begin{aligned}
&\mathbb{E}\left[\left\|Q(\eta \boldsymbol{g}^{(t)}) - \eta \boldsymbol{g}_e(\mathbf{w}^{(t)})\right\|^2\right] \\
&= \mathbb{E}\left[\left\|Q(\eta \boldsymbol{g}^{(t)}) - \eta \boldsymbol{g}^{(t)}\right\|^2\right] + \mathbb{E}\left[\left\|\eta \boldsymbol{g}^{(t)} - \eta \boldsymbol{g}_e(\mathbf{w}^{(t)})\right\|^2\right] \\
&\overset{(a)}{\leq} \gamma^2 \mathbb{E}\left[\left\|\eta \boldsymbol{g}^{(t)}\right\|^2\right] + \mathbb{E}\left[\left\|\eta \boldsymbol{g}^{(t)} - \eta \boldsymbol{g}_e(\mathbf{w}^{(t)})\right\|^2\right] \\
&= \gamma^2 \eta^2 \left[\mathbb{E}\left[\left\|\boldsymbol{g}^{(t)} - \boldsymbol{g}_e(\mathbf{w}^{(t)})\right\|^2\right] + \left\|\boldsymbol{g}_e(\mathbf{w}^{(t)})\right\|^2\right] + \eta^2 \mathbb{E}\left[\left\|\boldsymbol{g}^{(t)} - \boldsymbol{g}_e(\mathbf{w}^{(t)})\right\|^2\right] \\
&\overset{(b)}{\leq} (\gamma^2 + 1)\eta^2 \frac{\sigma^2}{bK} + \gamma^2 \eta^2 \left\|\boldsymbol{g}_e(\mathbf{w}^{(t)})\right\|^2,
\end{aligned}
$$

where (a) follows Assumption 3. (b) is due to Assumption 2 and the independence of mini-batch sampling noises among clients. □

With Lemma 1, we prove Theorem 2 as follows:

*Proof.* The model update in the $t$-th iteration can be expressed as $\mathbf{w}^{(t+1)} = \mathbf{w}^{(t)} - Q(\eta \boldsymbol{g}^{(t)})$. According to the Taylor's expansion, we have:

$$
\begin{aligned}
&\mathbb{E}\left[\ell(\mathbf{w}^{(t+1)})\right] - \mathbb{E}\left[\ell(\mathbf{w}^{(t)})\right] \\
&\leq -\mathbb{E}\left\langle \boldsymbol{g}_e(\mathbf{w}^{(t)}), Q(\eta \boldsymbol{g}^{(t)}) \right\rangle + \frac{L}{2}\mathbb{E}\left[\left\|Q(\eta \boldsymbol{g}^{(t)})\right\|^2\right] \\
&\overset{(c)}{=} -\mathbb{E}\left\langle \boldsymbol{g}_e(\mathbf{w}^{(t)}), \eta \boldsymbol{g}_e(\mathbf{w}^{(t)}) \right\rangle + \frac{L}{2}\mathbb{E}\left[\left\|Q(\eta \boldsymbol{g}^{(t)})\right\|^2\right] \\
&\overset{(d)}{=} -\eta \mathbb{E}\left[\|\boldsymbol{g}_e(\mathbf{w}^{(t)})\|^2\right] + \frac{L}{2}\mathbb{E}\left[\left\|Q(\eta \boldsymbol{g}^{(t)}) - \eta \boldsymbol{g}_e(\mathbf{w}^{(t)})\right\|^2\right] + \frac{L}{2}\mathbb{E}\left[\left\|\eta \boldsymbol{g}_e(\mathbf{w}^{(t)})\right\|^2\right] \\
&\overset{(e)}{\leq} -\left(\eta - \frac{\eta^2 L}{2}\right)\mathbb{E}\left[\|\boldsymbol{g}_e(\mathbf{w}^{(t)})\|^2\right] + \frac{L}{2}\left((\gamma^2 + 1)\eta^2 \frac{\sigma^2}{bK} + \gamma^2 \eta^2 \mathbb{E}\left[\|\boldsymbol{g}_e(\mathbf{w}^{(t)})\|^2\right]\right) \\
&= -\left(\eta - \frac{\eta^2 L}{2} - \frac{\eta^2 \gamma^2 L}{2}\right)\mathbb{E}\left[\|\boldsymbol{g}_e(\mathbf{w}^{(t)})\|^2\right] + \frac{\eta^2 L \sigma^2}{2bK}(\gamma^2 + 1),
\end{aligned}
$$

where (c) follows Lemma 1, (d) holds according to the fact that $\mathbb{E}\left\langle \nabla Q(\eta \boldsymbol{g}^{(t)}) - \eta \boldsymbol{g}_e(\mathbf{w}^{(t)}), \boldsymbol{g}_e(\mathbf{w}^{(t)}) \right\rangle = 0$, and (e) is due to Assumption 2-3. If it holds

Table 3: Details of the datasets

|  | MNIST | Fashon-MNIST | EMNIST | CIFAR-10 | CIFAR-100 | SVHN |
|---|---|---|---|---|---|---|
| No. of classes | 10 | 10 | 47 | 10 | 100 | 10 |
| No. of training samples | 60,000 | 60,000 | 112,800 | 50,000 | 50,000 | 73,257 |
| No. of test samples | 10,000 | 10,000 | 18,800 | 10,000 | 10,000 | 26,032 |
| Image size | $28 \times 28$ | $28 \times 28$ | $28 \times 28$ | $32 \times 32$ | $32 \times 32$ | $32 \times 32$ |
| License | Creative Commons Attribution-Share Alike 3.0 License | MIT License | Apache License 2.0 | MIT License | MIT License | CC0:Public Domain License |

Table 4: Hyperparameters for our DReS-FL method

| Parameters | MNIST | Fashion-MNIST | EMNIST | CIFAR-10 | CIFAR-100 | SVHN |
|---|---|---|---|---|---|---|
| Maximum L2-norm for gradient clipping | $2 \times 10^4$ | $2 \times 10^4$ | $5 \times 10^6$ | $2 \times 10^4$ | $1 \times 10^9$ | $2 \times 10^4$ |
| Prime number $p$ | $2^{200} - 75$ | $2^{200} - 75$ | $2^{440} - 33$ | $2^{440} - 33$ | $2^{440} - 33$ | $2^{440} - 33$ |
| Parameter $l$ in data transformation | 4 | 4 | 4 | 2 | 2 | 2 |

that $\eta - \frac{\eta^2 L}{2} - \frac{\eta^2 \gamma^2 L}{2} > 0$, we summarize the above inequality over $t = 1, 2, \ldots, \tau'$ to conclude the proof. $\qquad\square$

## B  Additional Experimental Details

All experiments are performed by Pytorch on an Intel Xeon Gold 6246R CPU @ 3.40 GHz and a Geforce RTX 3090. Some details of the datasets are summarized in Table 3. We adopt mini-batch SGD with a batch size of 64 to optimize the models in federated training. The communication round is set to be $7 \times 10^4$, and the clients perform one local SGD step in each round. The learning rate is initialized as 0.1, and it will decay with a factor of 0.65 after every 1500 rounds. Other parameters in our DRes-FL framework are summarized in Table 4.

## C  Complexity Analysis and Comparison

In this part, we analyze the communication and computational complexities of the proposed DReS-FL framework with respect to the parameters $(N, T, K, \tau, d_w, b_g)$. Parameter $N$ is the number of clients, and $T$ denotes the privacy threshold in Lagrange coding [22]. Parameter $K$ denotes the number of shards in the local datasets. A large value of $K$ reduces the communication and computation overheads in the proposed DReS-FL framework. In the federated training, the parameter $\tau$ corresponds to the number of communication rounds. Parameters $d_w$ and $b_g$ denote the model size and the global batch size, respectively. Before training starts, each client's computation cost for Lagrange coding and communication complexity for data sharing are $\mathcal{O}(N \log^2(K + T) \log \log(K + T))$ and $\mathcal{O}(N/K)$, respectively. In each round of federated training, the local computation complexity is $\mathcal{O}(d_w b_g / K)$, and the model uploading cost is $\mathcal{O}(d_w)$. Besides, the communication overhead of the server for model distributing is $\mathcal{O}(N d_w)$, and the model decoding complexity by polynomial interpolation is $\mathcal{O}(R \log^2 R \log \log R d_w)$, where $R$ denotes the minimum uploads needed for gradient decoding.

Different from our method, secure aggregation approaches [39, 53, 40, 42, 44, 41] generate random masks to protect the local model parameters. In each round, clients first share coded masks with each other, which allows for aggregating the masked models at the server. As some clients may drop out of the training process unexpectedly, the surviving clients upload the shared information belonging to the dropped clients to reconstruct the aggregated model. The main drawback of such approaches is that the clients need to generate new masks in each round, and their computational and communication complexities increase linearly with the number of training rounds. In comparison, the data sharing phase of our method only introduces extra costs for one time, which is independent of the training rounds. In the scenario that the number of training rounds is very large, the proposed DReS-FL method achieves lower computational and communication costs than the secure aggregation protocols.

Table 5: Computational complexity comparison

| | Preparation | Iterative training ($\tau$ rounds) | | |
|---|---|---|---|---|
| | Lagrangian coding | Generating coded random masks | Local model update | Global model aggregation |
| FedAvg | — | — | $\mathcal{O}\left(\tau d_w b_g/N\right)$ | $\mathcal{O}(\tau N d_w)$ |
| FedAvg with LightSecAgg | — | $\mathcal{O}\left(\frac{\tau d_w N^2 \log N}{R-T}\right)$ | $\mathcal{O}\left(\tau d_w b_g/N\right)$ | $\mathcal{O}\left(\frac{\tau d_w R \log R}{R-T}\right)$ |
| DReS-FL | $\mathcal{O}(N^2 \log^2(K+T)$ $\log\log(K+T))$ | — | $\mathcal{O}\left(\tau d_w b_g/K\right)$ | $\mathcal{O}(\tau d_w R \log^2 R$ $\log\log R)$ |

Table 6: Communication complexity comparison

| | Preparation | Iterative training ($\tau$ rounds) | | | |
|---|---|---|---|---|---|
| | Data sharing | Coded masks sharing among clients | Local model uploading | Coded masks uploading | Global model downloading |
| FedAvg | — | — | $\mathcal{O}(\tau d_w)$ | — | $\mathcal{O}(\tau N d_w)$ |
| FedAvg with LightSecAgg | — | $\mathcal{O}\left(\frac{\tau N^2 d_w}{R-T}\right)$ | $\mathcal{O}(\tau d_w)$ | $\mathcal{O}\left(\frac{\tau d_w R}{R-T}\right)$ | $\mathcal{O}(\tau N d_w)$ |
| DReS-FL | $\mathcal{O}(N^2/K)$ | — | $\mathcal{O}(\tau d_w)$ | — | $\mathcal{O}(\tau N d_w)$ |

The detailed comparisons among FedAvg, FedAvg with LighSecAgg [41], and our DReS-FL method are summarized in Table 5 and 6.

## D  Model Extension

Our DReS-FL framework can be extended to more general cases in which clients can run $s$ ($s \geq 1$) local SGD steps each round. Denote the computation results after $s$ local SGD steps in round $t$ as $\Delta \widetilde{\mathbf{w}}_j(s; \mathbf{w}^{(t)})$ for $j \in [N]$. Specifically, $\Delta \widetilde{\mathbf{w}}_j(s = 1; \mathbf{w}^{(t)}) \triangleq \widetilde{\boldsymbol{g}}(\widetilde{\mathbf{X}}_j^{(\mathcal{I}_t)}, \widetilde{\mathbf{Y}}_j^{(\mathcal{I}_t)}; \mathbf{w}^{(t)})$ and $\Delta \widetilde{\mathbf{w}}_j(s = 2; \mathbf{w}^{(t)}) \triangleq \widetilde{\boldsymbol{g}}(\widetilde{\mathbf{X}}_j^{(\mathcal{I}_{t+1})}, \widetilde{\mathbf{Y}}_j^{(\mathcal{I}_{t+1})}; \mathbf{w}^{(t)} - \frac{\eta}{bK}\widetilde{\boldsymbol{g}}(\widetilde{\mathbf{X}}_j^{(\mathcal{I}_t)}, \widetilde{\mathbf{Y}}_j^{(\mathcal{I}_t)}; \mathbf{w}^{(t)}))$. By carefully selecting the learning rate $\eta$ such that $\frac{\eta}{bK} \in \mathbb{F}_p$, the function $\Delta \widetilde{\mathbf{w}}_j(s; \mathbf{w}^{(t)})$ is still a polynomial in the finite field $\mathbb{F}_p$. Therefore, the central server can recover the desired model update by polynomial interpolation at the cost of low dropout-resiliency caused by the high degree of $\Delta \widetilde{\mathbf{w}}_j(s; \mathbf{w}^{(t)})$.

## E  Degree of Gradient in PINN

Given a data sample as $(\boldsymbol{x}, \boldsymbol{y})$, the feedforward process of a PINN with $L$ quadratic activation layers $\boldsymbol{h}$ is as follows:

$$\boldsymbol{z}_0' \rightarrow \boldsymbol{z}_1 \rightarrow \boldsymbol{z}_1' \rightarrow \boldsymbol{z}_2 \rightarrow \boldsymbol{z}_2' \rightarrow \cdots \rightarrow \boldsymbol{z}_L \rightarrow \boldsymbol{z}_L' \rightarrow \boldsymbol{z}_{L+1},$$

where $\boldsymbol{z}_0' \triangleq \boldsymbol{x}$, $\boldsymbol{z}_l' = \boldsymbol{h}(\boldsymbol{z}_l)$, and $\boldsymbol{z}_l = \mathbf{W}_l \boldsymbol{z}_{l-1}' + \boldsymbol{b}_l$ for $l \in 1 : L+1$. $\boldsymbol{z}_{L+1}$ is the output of PINN, and the loss function is the squared error between $\boldsymbol{y}$ and $\boldsymbol{z}_{L+1}$, i.e., $\ell = \|\boldsymbol{y} - \boldsymbol{z}_{L+1}\|_2^2$. The notations $\mathbf{W}_l$ and $\boldsymbol{b}_l$ correspond to the weight matrix and bias vector in PINN. With the above preparation, we derive the gradients as follows:

$$\frac{\partial \ell}{\partial \boldsymbol{z}_{L+1}} = 2(\boldsymbol{z}_{L+1} - \boldsymbol{y})^T, \ \frac{\partial \boldsymbol{z}_l'}{\partial \boldsymbol{z}_l} = 2\text{diag}(\boldsymbol{z}_l), \ \frac{\partial \boldsymbol{z}_l}{\partial \mathbf{W}_l[j]} = \mathbf{I}\boldsymbol{z}_{l-1}'[j], \ \frac{\partial \boldsymbol{z}_l}{\partial \boldsymbol{z}_{l-1}} = \mathbf{W}_l,$$

where $\mathbf{W}_l[j]$ is the $j$-th column in $\mathbf{W}_l$, and $\boldsymbol{z}_{l-1}'[j]$ is the $j$-th element in the vector $\boldsymbol{z}_{l-1}'$. According to the chain rule of gradient, the gradient of the loss function with respect to the weight $\mathbf{W}_l[j]$ is

$$\begin{aligned}
\frac{\partial \ell}{\partial \mathbf{W}_l[j]} &= \frac{\partial \ell}{\partial \boldsymbol{z}_{L+1}} \frac{\partial \boldsymbol{z}_{L+1}}{\partial \boldsymbol{z}_L'} \frac{\partial \boldsymbol{z}_L'}{\partial \boldsymbol{z}_L} \cdots \frac{\partial \boldsymbol{z}_{l+1}}{\partial \boldsymbol{z}_l'} \frac{\partial \boldsymbol{z}_l'}{\partial \boldsymbol{z}_l} \frac{\partial \boldsymbol{z}_l}{\partial \mathbf{W}_l[j]}, \\
&= 2(\boldsymbol{z}_{L+1} - \boldsymbol{y})^T \mathbf{W}_{L+1} 2\text{diag}(\boldsymbol{z}_L) \cdots \mathbf{W}_{l+1} 2\text{diag}(\boldsymbol{z}_l) \boldsymbol{z}_{l-1}'[j].
\end{aligned}$$

Note that the mappings from input $z'_0$ to $z'_l$ and $z_{l+1}$ are polynomials with degree $2^l$. Therefore, the degree of the gradient $\frac{\partial \ell}{\partial \mathbf{W}_l[j]}$ is

$$\deg\left(\frac{\partial \ell}{\partial \mathbf{W}_l[j]}\right) = 2^L + 2^{L-1} + \cdots + 2^{l-1} + 2^{l-1},$$
$$= 2^{L+1}.$$