# OpenReview forum: "DReS-FL: Dropout-Resilient Secure Federated Learning for Non-IID Clients via Secret Data Sharing"
_NeurIPS.cc/2022/Conference — NeurIPS 2022 Accept_

### Official Review · Reviewer_kTxQ · 2022-07-10

**Rating:** 4
**Confidence:** 4
**Soundness:** 2 fair
**Presentation:** 2 fair
**Contribution:** 2 fair

**Summary:**

This paper proposes a new framework named Dropout-Resilient Secure Federated Learning (DReS-FL) to address the non-iid and client dropout issues in federated learning. By leveraging Lagrange coded computing (LCC) and secret sharing, DReS-FL enables clients to securely share their data to build a ‘public data set’ and participate in the learning task. To further guarantee the gradients are polynomial functions in a finite field, this paper introduces the polynomial integer neural networks (PINNs) to enable the framework. The experiment results show the DReS-FL proposed can achieve higher accuracy when facing clients dropping out.


**Questions:**

1.	To perform Secret Sharing, the parameter K should be set before the learning task start, but whether K is constant?
2.	Line 156 said a large value of K can reduce the communication cost, can you explain the reason? (as I think more shards may bring more communication times between different clients)
3.	How can the clients establish communication?
4.	Are there any rules for sharing the data? How can a client determine whom he should share the shards with?


**Limitations:**

See the questions.
As the communication cost is often a bottleneck for FL, involving massive communication may further worsen this issue; please consider providing the discussion.


**Strengths And Weaknesses:**

Strengths:
1.	Addressing the FL dropout issue is a hot topic, and the proposed framework is very interesting.
2.	This paper proposes the polynomial integer neural networks (PINNs) and proves PINNs can ensure the correct decode of the global gradient without privacy leakage.
3.	The DReS-FL has been formally analyzed to be resilient to a certain number N of client dropouts.

Weakness:
The secret sharing mechanism is not well presented.

---

> ### Author Response · Authors · 2022-08-02
> **Response to Reviewer kTxQ (Part 2)**
>
> > $\textbf{Question 5:}$ As the communication cost is often a bottleneck for FL, involving massive communication may further worsen this issue; please consider providing the discussion.
>
> $\textbf{Answer 5:}$ Thank you for this suggestion. Our method achieves comparable or even lower communication overhead compared with the secure aggregation methods while providing the same security guarantee. In the following, we specify the communication complexities of the proposed DRes-FL method, the secure sharing methods, as well as the FedAvg method for comparison. More details about the computational and communication complexities had been discussed in Section C of Appendix.
>
> The communication complexity of the proposed DReS-FL framework is analyzed with respect to the parameters $(N,T,K,\tau,d_{w},b_{g})$. Parameter $N$ is the number of clients, and $T$ is a privacy parameter in Lagrange coding [1]. Parameter $K$ denotes the number of shards in the local datasets. In federated training, the parameter $\tau$ corresponds to the number of training rounds. Parameters $d_{w}$ and $b_{g}$ denote the model size and the global batch size, respectively. Before training starts, each client's communication complexity for data sharing is $\mathcal{O}(N / K)$. In each round of federated training, the model uploading cost is $\mathcal{O}(d_{w})$, and the communication overhead of the server for model distributing is $\mathcal{O}(Nd_{w})$. Different from our method, secure aggregation approaches (e.g., LightSecAgg [2]) generate random masks to protect the local model parameters. In each round, clients first share coded masks with each other, which allows for aggregating the masked models at the server. As some clients may drop out of the training process unexpectedly, the surviving clients upload the shared information belonging to the dropout clients to reconstruct the aggregated model. The complexity comparison among FedAvg, FedAvg with LightSecAgg [2], and the proposed DReS-FL method are summarized in the following table, where the parameter $R$ denotes the minimum uploads needed for gradient decoding.
>
> Table 1: Communication complexity comparison
>
>  |                         | Data sharing (before training starts) | Coded masks sharing among clients ($\tau$ training rounds) | Local model uploading ($\tau$ training rounds) |    Coded masks uploading ($\tau$ training rounds)    | Global model downloading ($\tau$ training rounds) |
>  | :---------------------: | :-----------------------------------: | :--------------------------------------------------------: | :--------------------------------------------: | :--------------------------------------------------: | :-----------------------------------------------: |
>  |         FedAvg          |                  $-$                  |                            $-$                             |           $\mathcal{O}(\tau  d_{w})$           |                         $-$                          |            $\mathcal{O}(\tau N d_{w})$            |
>  | FedAvg with LightSecAgg |                  $-$                  |  $\mathcal{O}\left( \frac{\tau N^{2}d_{w}}{R-T} \right)$   |           $\mathcal{O}(\tau  d_{w})$           | $\mathcal{O}\left( \frac{\tau d_{w} R}{R-T}\right) $ |            $\mathcal{O}(\tau N d_{w})$            |
>  |         DReS-FL         |        $\mathcal{O}(N^{2}/K)$         |                            $-$                             |           $\mathcal{O}(\tau  d_{w})$           |                         $-$                          |            $\mathcal{O}(\tau N d_{w})$            |
>
>  This table shows that the communication costs of both the secure aggregation method and our DReS-FL method increase quadratically with the number of clients $N$. But, the main drawback of the secure aggregation methods is that the clients need to generate and share new masks in each round, which results in high communication overhead when the number of training rounds $\tau$ is very large. In comparison, our method only introduces extra costs for one time in secret data sharing before the training starts. When the training round $\tau$ is very large, the proposed DReS-FL method achieves lower communication costs compared with the secure aggregation protocols.
>
> Overall, as the communication overhead introduced by the secret data sharing is a one-time cost, the authors believe the complexity of our method is acceptable.
>
> [1] Yu et al. "Lagrange coded computing: Optimal design for resiliency, security, and privacy," AISTATS, 2019.
>
> [2] So et al. "Lightsecagg: a lightweight and versatile design for secure aggregation in federated learning," MLSys, 2022.

---

> ### Author Response · Authors · 2022-08-02
> **Response to Reviewer kTxQ (Part 1)**
>
> We thank the reviewer for the time and effort spent on our manuscript. We have revised the manuscript and provided a notation table to help the reviewer to better understand the proposed method.  The weaknesses and comments are itemized in the following, and we answer them accordingly.
>
> > $\textbf{Question 1:}$ To perform Secret Sharing, the parameter $K$ should be set before the learning task start, but whether $K$ is constant?
>
> $\textbf{Answer 1:}$ Thank you for this comment. The parameter $K$ is unchanged during the training process.
>
> > $\textbf{Question 2:}$ Line 156 said a large value of $K$ can reduce the communication cost, can you explain the reason? (as I think more shards may bring more communication times between different clients)
>
> $\textbf{Answer 2:}$ As discussed in Section C of Appendix, the communication complexity in the data sharing phase is $\mathcal{O}\left(N^{2} / K\right)$, where $N$ is the number of clients. Therefore, increasing the value of $K$ reduces the communication cost. The detailed explanation is as follows:
>
> First, at each client $i \in [N]$, the local dataset $(\overline{\mathbf{X}}^{(i)},\overline{\mathbf{Y}}^{(i)})$ is partitioned to $K$ shards as $ \overline{\mathbf{X}}^{(i)} \triangleq [\overline{\mathbf{X}}\_{1}^{(i)T}, \ldots,\overline{\mathbf{X}}\_{K}^{(i)T}]^{T}$ and $ \overline{\mathbf{Y}}^{(i)} \triangleq [\overline{\mathbf{Y}}\_{1}^{(i)T}, \ldots,\overline{\mathbf{Y}}\_{K}^{(i)T}]^{T}$, where $\overline{\mathbf{X}}\_{k}^{(i)} \in \mathbb{F}\_{p}^{\frac{m\_{i}}{K} \times d\_{x}}$ and $\overline{\mathbf{Y}}\_{k}^{(i)} \in \mathbb{F}\_{p}^{\frac{m\_{i}}{K} \times d\_{y}}$. The parameter $m_{i}$ denotes the number of samples at client $i$. $d_{x}$ and $d_{y}$ correspond to the dimensions of the input variable and the output vector, respectively. Then, as introduced in Section 4.1 of the manuscript, the client $i \in [N]$ uses polynomial functions $\mathbf{u}\_{i}: \mathbb{F}\_{p} \rightarrow \mathbb{F}\_{p}^{\frac{m_{i}}{K}\times d_{x}}$ and $\mathbf{v}\_{i}: \mathbb{F}\_{p} \rightarrow \mathbb{F}\_{p}^{\frac{m_{i}}{K}\times d_{y}}$ to obtain $N$ encoded datasets $(\widetilde{\mathbf{X}}\_{j}^{(i)},\widetilde{\mathbf{Y}}\_{j}^{(i)}) \triangleq (\mathbf{u}\_{i}(\alpha_{j}), \mathbf{v}\_{i}(\alpha_{j}))$ for $j \in [N]$, where each encoded dataset $(\widetilde{\mathbf{X}}\_{j}^{(i)},\widetilde{\mathbf{Y}}\_{j}^{(i)})$ is sent to client $j \in [N] \backslash\{i\}$ from client $i$. As $\widetilde{\mathbf{X}}\_{j}^{(i)}$ is a $ \frac{m_{i}}{K} \times d_{x}$ matrix, and $\widetilde{\mathbf{Y}}\_{j}^{(i)}$ is a $ \frac{m_{i}}{K} \times d_{y}$ matrix, the amount of communication overhead in secret data sharing is proportional to $\frac{\sum_{i=1}^{N}m_{i}(d_{x}+d_{y})(N-1)N}{K}$, which is negatively related to $K$.
>
> > $\textbf{Question 3:}$ How can the clients establish communication?
>
> $\textbf{Answer 3:}$ Thank you for this comment. In the federated learning system, communication among clients can be established by using the central server as a relay node. Public-key cryptography can be used for encrypting communication to provide confidentiality.
>
> > $\textbf{Question 4:}$ Are there any rules for sharing the data? How can a client determine whom he should share the shards with?
>
> $\textbf{Answer 4:}$ As mentioned in $\textbf{Answer 2}$, every client will send the encoded datasets to all the other clients.

---

> ### Author Response · Authors · 2022-08-08
> **Please let us know if anything else is still unclear.**
>
> Dear __Reviewer kTxQ__,
>
> Thank you for your time spent reviewing our manuscript. We have addressed all your concerns in the following posts. Could you please let us know if you have any other questions about our work? We are gladly willing to answer your further questions.
>
> Thanks,
>
> Paper6370 Authors

---

### Official Review · Reviewer_gqjh · 2022-07-11

**Rating:** 6
**Confidence:** 4
**Soundness:** 2 fair
**Presentation:** 4 excellent
**Contribution:** 3 good

**Summary:**

This paper aims to address the non-iid and dropout problems in (privacy-preserving) FL using the Lagrange coded computing framework. Apply the LCC framework, it proposes a class of network called Polynomial Integer Neural Networks.


**Questions:**

(1) How are the baseline setups different than the main method (i.e., how do they tackle non-iid, dropout problems)?
(2) The author claims "We observe that FedAvg is influenced by the dropout problem due to the biased gradients" (in lines 295-296). How is this inferred from Table 1 and figure 2?

Overall, the paper is well written and shows enough technical strengths. Please address the questions in experiment setups and analysis. I am willing to raise my score if the experiments are good as well.


**Limitations:**

The author has stated the limitation in the main paper.

**Strengths And Weaknesses:**

Strengths:
    (1) The paper is clearly written - problems, motivations, and intuition of the main methods are easy to understand.
    (2) The method is novel and original (the paper shows how to use PINN to adopt LCC in the setting).
    (3) Theoretical guarantees are given.
    (4) The limitation is stated - the degree of the gradient grows fast so that the network can not be deep, which is reasonable.

Weakness:
    (1) The experiment setup for baselines is not clearly stated - How is the FedAvg setup? Is it combined with secure aggregation methods?
    (2) The experiment results are not clearly explained. Please see below for concrete questions.

---

> ### Author Response · Authors · 2022-08-02
> **Response to Reviewer gqjh**
>
> We thank the reviewer for the positive feedback. The weaknesses and comments are itemized in the following, and we answer them accordingly.
>
> > $\textbf{Question 1:}$ The experiment setup for baselines is not clearly stated - How is the FedAvg setup? Is it combined with secure aggregation methods?
>
> $\textbf{Answer 1:}$ The baseline methods, including FedAVG, FedAVG-IS, and SCAFFOLD, are not combined with secure aggregation methods. The objective of the experiments is to investigate the negative effect of data heterogeneity and the dropout problem on convergence performance. As the quantization step in secure aggregation may further degrade the convergence performance, the secure aggregation methods have not been applied in the experiments.
>
> > $\textbf{Question 2.1:}$ How are the baseline setups different than the main method (i.e., how do they tackle non-iid, dropout problems)?
>
> $\textbf{Answer 2.1:}$ Thank you for this comment. We add more details about the baseline methods as follows:
>
> (1) FedAvg: At each communication round, the server simply aggregates the local updates from the survival clients. This method induces bias when the local data distribution and dropout probability vary among devices.
>
> (2) FedAvg with importance sampling (FedAvg-IS): Assume the server knows the dropout distribution. Each local update from the survival clients is weighted by the participation probability (i.e., 1 - dropout probability). This approach can solve the dropout problem if the local datasets are IID.
>
> (3) SCAFFOLD: This method introduces control variates to correct for the "client-drift" in its local updates, which is robust to the data heterogeneity and the client sampling. However, when the clients frequently drop out of the training process beyond the control of the central server, the SCAFFOLD method suffers performance degradation.
>
> (4) Centralized training: The server can access all the clients' datasets and uses them for model training.
>
> In the revision, the details of the baselines have been described in Appendix A.
>
> > $\textbf{Question 2.2:}$ The author claims "We observe that FedAvg is influenced by the dropout problem due to the biased gradients". How is this inferred from Table 2 and Figure 2?
>
> $\textbf{Answer 2.2:}$ Sorry for the confusion. What we meant is that the FedAvg method achieves lower accuracy compared with the centralized training scheme as shown in Table 2. This is attributed to the non-IID data and client dropouts. We have revised the manuscript for better clarity.

---

> > ### Comment · Reviewer_gqjh · 2022-08-09
> > **Thank you for the response**
> >
> > After carefully reading the rebuttal, my major concerns have been addressed by the authors' responses. I will thus increase my overall evaluation score from 5 to 6.

---

> ### Author Response · Authors · 2022-08-08
> **Please let us know whether we addressed your concerns**
>
> Dear __Reviewer gqjh__,
>
> We have answered your questions in experiment setups and analysis. Can we kindly ask you to see whether the raised concerns have been well addressed? We are willing to hear from you again.
>
> Thank you,
>
> Paper6370 Authors

---

### Official Review · Reviewer_FXUt · 2022-07-11

**Rating:** 7
**Confidence:** 4
**Soundness:** 3 good
**Presentation:** 4 excellent
**Contribution:** 3 good

**Summary:**

This paper proposes a Dropout-Resilient Secure Federated Learning framework based on Lagrange coded computing to
tackle two problems in FL: the non-IID data distribution and client dropout.

The clients share their datasets secretly using LCC to handle the non-IID problem and client dropouts. At the beginning of the training, the clients secretly share their private datasets with each other via Lagrange coding. This allows clients to access an encoded version of the global dataset. After collecting the computation results from a certain number of surviving clients, the server performs polynomial interpolation to decode the gradient.

The authors introduce the idea of polynomial integer neural networks (PINNs) for their proposed framework to work, since they need to correctly decode the gradient at the server, and that the gradient has to be a polynomial function in a finite field.

The authors provide some theoretical analyses and numerical experiments to back their claims.

**Questions:**

Have the authors also looked the LEAF datasets for the experiments? LEAF seems to be the standard dataset for FL papers, with natural client partitions and other FL constraints.

**Limitations:**

As pointed out by the authors,

- the generality of the idea is questionable. Requiring neural networks to be polynomial integer neural networks (PINNs) is an assumption that probably cannot be hold in practice.
- Similarly, the degree of the gradientin a PINN increases exponentially with the number of layers of the neural network, limiting training larger models
- multiple local SGD steps is yet not supported, which is quite common in practical FL settings

**Strengths And Weaknesses:**

Strengths:
- The paper is well written and it is very easy to read
- The paper is back by theoretical analyses and also backed by numerical experiments
- The idea of the paper is quite novel and interesting, even though it may not be extensible to big and practical neural networks and practical FL settings


Weaknesses:
- As correctly pointed out by the authors, the generality of the idea is questionable. Requiring neural networks to be polynomial integer neural networks (PINNs) is an assumption that probably cannot be hold in practice.

---

> ### Author Response · Authors · 2022-08-02
> **Response to Reviewer FXUt**
>
> We thank the reviewer for the positive feedback. Our work focuses on tackling both the non-IID and dropout problems in federated learning. The key idea is to utilize Lagrange coding to secretly share the private datasets among clients so that the effects of non-IID distribution and client dropouts can be compensated during local gradient computations. Theoretical analysis shows that the proposed DReS-FL method is robust to data heterogeneity and client dropouts while providing privacy protection for the local datasets. For the practical implementation, we agree with the reviewer that our proposed method has several constraints. We will try to solve these challenging problems in the future.
>
> > $\textbf{Question1:}$ Have the authors also looked the LEAF datasets for the experiments? LEAF seems to be the standard dataset for FL papers, with natural client partitions and other FL constraints.
>
> $\textbf{Answer 1:}$ Thank you for pointing out the valuable LEAF datasets, we will consider using these datasets in our future works.

---

> > ### Comment · Reviewer_FXUt · 2022-08-08
> > **Response to the authors**
> >
> > Given the response of the authors, I decide to keep the score. Thank you!

---

### Official Review · Reviewer_ULSx · 2022-07-17

**Rating:** 4
**Confidence:** 4
**Soundness:** 2 fair
**Presentation:** 2 fair
**Contribution:** 2 fair

**Summary:**

The paper considers secure federated learning (FL) and aims to tackle the challenges of non-IID data distribution across clients and dropouts. The paper proposes that each client first secret shares their data to all other clients using Lagrange Coded Computing. This involves quantizing the data and appropriately mapping it to a finite field. Then, gradients are computed over the secret shared data. Since secret sharing operates over a finite field (as opposed to real numbers), the paper proposes to use *polynomial integer neural networks*, where quadratic function is used as a non-linearity (instead of ReLU or other typical functions) and a mean squared loss is used (instead of cross-entropy loss). The server first ‘decodes’ the gradients computed on secret shared data to obtain the gradient on the quantized data. Then, it sums the gradients and takes a step to update the global model.

**Questions:**

1. The key technical idea in the paper involves computing gradients on secret shared data. However, the paper does not give details on the gradient computation on secret shared data. It simply mentions that the gradient (when loss is MSE and non-linearity is square function) would be a multivariate polynomial in $(\bar{X}, \bar{Y})$ and parameters $\mathbf{w}$. Without any details, it is difficult to understand how the server decodes $\tilde{g}(\tilde{X}_j^{(I_t)}, \tilde{Y}_j^{(I_t)})$ in line 9 of Algorithm 1. During secret sharing, all the $K$ shards of each client are encoded together, while the server decodes gradients for each shard separately. It is important to add details and provide a proof of how the decoding happens.
Currently, it is not even clear how the gradient $\tilde{g}(\tilde{X}_j^{(I_t)}, \tilde{Y}_j^{(I_t)})$ is defined. Is this an average of the gradients in  batch $I_t$?

2. In Theorem 2, the security is mentioned in terms of  $\tilde{g}(\tilde{X}_j^{I_t}, \tilde{Y}_j^{I_t})$. However, the server decodes the gradients to obtain the gradient on uncoded data $\tilde{g}(\bar{X}_j^{(I_t)}, \bar{Y}_j^{(I_t)})$ for $j\in[K]$. Is it not that the privacy for server should be proved in terms of the decoded gradient? In other words, Theorem 2 should be stated in terms of $\tilde{g}(\bar{X}_j^{(I_t)}, \bar{Y}_j^{(I_t)})$? Overall, it is not clear how the server only obtains an *aggregation* of gradients. It is important to expand on this.

3.  In FL all communication typically happens through the server. It is important to mention how a client will securely communicate the secret shared data to all other clients.

4. The proposed scheme involves several approximations as stated below:
* Quantization to convert real-field data into integers
* Square function as a non-linearity instead of ReLU
* Mean squared loss instead of cross-entropy loss
* Stochastic quantization during update (eq. (6))
* Neural network weights are restricted to be integers

It seems like these approximations will negatively impact the accuracy. However, performance evaluation in Fig. 2 depicts that the proposed method consistently outperforms FedAvg for all datasets. It would be important to discuss why these factors mentioned above do not impact accuracy, and if they do, then are any optimizations performed to overcome them. Otherwise, the results appear to be a bit 'magical'.

In particular, it is known that the cross-entropy loss typically performs better than MSE. Furthermore, quadratic approximation for ReLU is known to severely degrade accuracy in some settings, see e.g., Mishra et al., “DELPHI: A Cryptographic Inference Service for Neural Networks,” Usenix 2020. Why these issues are not observed in the proposed method, and if they are observed how they are overcome?

5. Convergence analysis considers population loss instead of empirical loss. Convergence analysis often considers empirical risk. It would be helpful to comment on this.

6. To tackle non-iid nature, there are solutions such as FedProx, which change the local optimization problem at the client side, but still perform averaging at the server side. It is straightforward to use secure aggregation with FedProx as one solution of privacy + data heterogeneity. How would the proposed method compare against FedProx + secure aggregation (or similar solutions)?

7. It would be helpful to specify if the paper focuses on cross-silo setup or cross-device setup. Since cross-device setup consists of a large number of clients, performing secret sharing across all clients may be costly. On the other hand, the problem of dropouts is less severe in the cross-silo setup. It would be good to add a discussion on this.

8. Footnote on page 4 says that “The constant scalar c could be the absolute value of the minimum entry in dataset (X, Y).“ Since the dataset $(X,Y)$ is distributed across clients, how is it possible to find such a scalar $c$? Also, if the value of $c$ is data dependent it may not be appropriate to call it a ‘constant’.

9. What field size $p$ is used in the experiments?

10. Experiments use MLP with two hidden layers with $K=1$ and $T=1$. It is not clear why the degree of gradient as a polynomial would be 8. Can the authors give more details?


**Limitations:**

As mentioned in the Weaknesses, the proposed solution would incur heavy costs during the secret sharing phase. Also, dropouts cannot be tolerated during secret sharing. In addition, as mentioned in Questions, the proposed scheme relies on several ‘approximations’, and it would be important to discuss their impact on the accuracy and/or speed of convergence.

**Strengths And Weaknesses:**

*Strengths:* The paper considers an important problem of tackling data heterogeneity and privacy in FL. These issues are intellectually challenging and practically important.

*Weaknesses:*

1. The proposed solution requires each client to secret share their data to all other clients. Clearly, this is going to incur substantial burden in terms of computation and communication costs. The paper does not explicitly characterize the computation and communication costs for secret sharing client data. (It just mentions that the ‘sharing’ parameter $K$ impacts the costs, but explicit analysis seems to be missing.)

2. One of the main motivations of the paper is to tackle client dropouts. However, it looks like all clients need to be present during the phase of secret sharing data. This assumption needs to be clearly stated and it would be good to add a discussion around this. In particular, if the communication cost of transferring secret shared data to other clients is large, then this phase will take considerable time. In this case, not allowing any dropouts during secret sharing and requiring all clients to be present can be a major limitation.

3. The paper does not give sufficient details in terms of gradient computation and securely transferring secret shared data to clients. (Specific comments in the Questions section.)

---

> ### Author Response · Authors · 2022-08-02
> **Response to Reviewer ULSx (Part 6)**
>
> > $\textbf{Question 8:}$ Footnote on page 4 says that "The constant scalar $c$ could be the absolute value of the minimum entry in the dataset $\left(\mathbf{X}, \mathbf{Y}\right)$." Since the datasets are distributed across clients, how is it possible to find such a scalar?
> Also, if the value of $c$ is data dependent it may not be appropriate to call it a "constant".
>
> $\textbf{Answer 8:}$ We would like to first clarify that the objective of adding a variable $c$ is to make sure that the transformed dataset $(\overline{\mathbf{X}}, \overline{\mathbf{Y}})$ is in a finite field after the transformation $\overline{\mathbf{X}} \triangleq {Round}\left(2^{l} \cdot \phi(\mathbf{X})\right)$, where $\phi(z)=z+c$. When the scalar $c$ is sufficiently large, each entry in the dataset $(\overline{\mathbf{X}}, \overline{\mathbf{Y}})$ is a positive integer. In practice, the authors believe it is easy to find a suitable scalar $c$ according to the input data format and/or the federated learning task. For example, if the input data are images in RGB format, we know that the minimum pixel value is 0.
>
> > $\textbf{Question 9:}$ What field size $p$ is used in the experiments?
>
> $ \textbf{Answer 9:}$ Please note that we had shown the prime number $p$ in Table 2 of Appendix.
>
>  |      |     MNIST      | Fashion-MNIST  |     EMNIST      |    CIFAR-10    |    CIFAR-100    |      SVHN      |
>  | :--: | :------------: | :------------: | :-------------: | :------------: | :-------------: | :------------: |
>  | $p$  | $10^{31} + 33$ | $10^{31} + 33$ | $10^{51} + 121$ | $10^{31} + 33$ | $10^{71} + 273$ | $10^{31} + 33$ |
>
> > $\textbf{Question 10:}$. Experiments use MLP with two hidden layers with $K=1$ and $T=1$. It is not clear why the degree of gradient as a polynomial would be 8. Can the authors give more details?
>
> $\textbf{Answer 10:}$ We would like to clarify that the degree of the gradient is independent of the parameters $K$ and $T$, and it is only determined by the number of layers in the deep learning model. Denote the number of hidden layers in a multi-layer perceptron (MLP) as $L$, the degree is $2^{L+1}$.
>
> We next provide an example to explain why the degree of the gradient is 8 in the experiment. Consider a regression problem with input variable $x\in \mathbb{R}^{1}$ and output variable $y \in \mathbb{R}^{1}$, and select an MLP with two hidden layers to learn the mapping from $x$ to $y$. Specifically, the hidden unit is set to 1 in each layer, and the additive biases are removed for simplicity. The weights in this MLP are denoted as $\mathbf{w}\_{1}, \mathbf{w}\_{2}, \mathbf{w}\_{3} \in \mathbb{R}^{1\times 1}$. Using the quadratic function as the activation function and adopting the mean squared error (MSE) to measure the distortion, the loss function is $\ell(\mathbf{w}\_{1},\mathbf{w}\_{2},\mathbf{w}\_{3}) = \\{y - \mathbf{w}\_{3}[\mathbf{w}\_{2}(\mathbf{w}\_{1}x)^{2}]^{2}\\}^{2}$. The partial derivative of the loss function $\ell(\mathbf{w}\_{1},\mathbf{w}\_{2},\mathbf{w}\_{3})$ with respect to $\mathbf{w}\_{1}$ is $\nabla_{\mathbf{w}\_{1}}\ell = 8\mathbf{w}\_{3}^{2}\mathbf{w}\_{2}^{4}\mathbf{w}\_{1}^{7}x^{8} - 8\mathbf{w}\_{3}\mathbf{w}\_{2}^{2}\mathbf{w}\_{1}^{3}x^{4}y $, which is a multivariate polynomial function of $(x,y)$, and its degree is 8.
>
> ** **
>
> **Overall, thank you very much for your valuable comments that improve the quality of our work. We look forward to hearing from you again.**

---

> ### Author Response · Authors · 2022-08-02
> **Response to Reviewer ULSx (Part 5)**
>
> > $\textbf{Question 4.2:}$ Quadratic approximation for ReLU is known to severely degrade accuracy in some settings, see e.g., Mishra et al., "DELPHI: A Cryptographic Inference Service for Neural Networks," Usenix 2020. Why these issues are not observed in the proposed method, and if they are observed how they are overcome?
>
> $\textbf{Answer 4.2:}$ The work DELPHI focuses on secure inference, which replaces ReLU activation in a pre-trained model with quadratic approximation to reduce the cost of non-linear operations. However, simply replacing the activation functions without fine-tuning severely degrades the model performance, since the output distribution of the activation function is largely changed. Our proposed method is different from DELPHI, where the neural network is trained with the quadratic function as the activation function from the scratch. The model parameters can be adaptively optimized according to the activation function in the end-to-end training process, which maintains the model performance.
>
> > $\textbf{Question 5:}$ Convergence analysis considers population loss instead of empirical loss. Convergence analysis often considers empirical risk. It would be helpful to comment on this.
>
> $\textbf{Answer 5:}$ Sorry for the confusion. Our convergence analysis indeed considers the empirical risk. We formulate the empirical loss function as $\ell(\mathbf{w}) \triangleq \mathbb{E}\_{(\overline{\mathbf{X}},\overline{\mathbf{Y}}) \sim \overline{\mathcal{D}}} \|\overline{\mathbf{Y}}-\boldsymbol{f}(\overline{\mathbf{X}} ; \mathbf{w})\|_{2}^{2}$ and define the gradient as $\boldsymbol{g}\_{e}\left(\mathbf{w}\right) \triangleq \mathbb{E}\_{(\overline{\mathbf{X}},\overline{\mathbf{Y}}) \sim \overline{\mathcal{D}}}[\boldsymbol{g}(\overline{\mathbf{X}},\overline{\mathbf{Y}};\mathbf{w})]$, where the variables $(\overline{\mathbf{X}},\overline{\mathbf{Y}})$ are drawn from the distribution of the global dataset $\overline{\mathcal{D}}$ (i.e., the concatenation of the clients’ training datasets $(\overline{\mathbf{X}}^{(i)}, \overline{\mathbf{Y}}^{(i)})$ for $i \in [N]$).
>
> > $\textbf{Question 6:}$ To tackle non-iid nature, there are solutions such as FedProx, which change the local optimization problem at the client side, but still perform averaging at the server side. It is straightforward to use secure aggregation with FedProx as one solution for privacy + data heterogeneity. How would the proposed method compare against FedProx + secure aggregation (or similar solutions)?
>
> $\textbf{Answer 6:}$ Thank you for the suggestion. We agree with the reviewer that there are many solutions that can alleviate the non-IID problem in federated learning and can be combined with secure aggregation methods. Nevertheless, the performance of these methods cannot perform well when the clients frequently drop out from the learning system, as it increases statistical heterogeneity in FL, which adversely impacts convergence behavior. In the manuscript, we selected a representative method, SCAFFOLD, as a baseline for comparison. The empirical results in Table 2 show that the SCAFFOLD method achieves the worst performance among the baselines. Our research is mainly motivated by a lack of effective methods that can simultaneously handle both non-IID data and client dropouts while preserving privacy.
>
> Besides, the reason we did not select FedProx as a baseline is that the dropout problem considered in FedProx is different from our work. The main contribution of FedProx is that it develops an effective way to leverage the "partial works" from the stragglers (if they cannot complete the local model update on time) rather than simply dropping them. In our work, we consider the scenario where the clients may abort the training halfway (e.g., due to battery level), which is out of the control of the central server.
>
> > $\textbf{Question 7:}$ It would be helpful to specify if the paper focuses on cross-silo setup or cross-device setup. Since cross-device setup consists of a large number of clients, performing secret sharing across all clients may be costly. On the other hand, the problem of dropouts is less severe in the cross-silo setup. It would be good to add a discussion on this.
>
> $\textbf{Answer 7:}$ Thank you for this comment. Our method can be implemented in the cross-silo setup and the cross-device setup since data heterogeneity is a general problem in federated learning. Besides, our method has the ability to tackle the dropout problem. The authors agree with the reviewer that a large number of clients in the cross-device setup increases the communication overhead during the data sharing phase, but, as mentioned in $\textbf{Response 1}$ (in $\textbf{Part 1}$ and $\textbf{Part 2}$), the authors believe the complexity of our method is still tolerable.

---

> ### Author Response · Authors · 2022-08-02
> **Response to Reviewer ULSx (Part 4)**
>
> > $\textbf{Question 2:}$ In Theorem 2, the security is mentioned in terms of $\widetilde{\boldsymbol{g}}(\widetilde{\mathbf{X}}\_{j}^{(\mathcal{I}\_{t})}, \widetilde{\mathbf{Y}}\_{j}^{\left(\mathcal{I}\_{t}\right)} ; \mathbf{w}^{(t)})$. However, the server decodes the gradients to obtain the gradient on uncoded data $\widetilde{\boldsymbol{g}}(\overline{\mathbf{X}}\_{k}^{(\mathcal{I}\_{t})}, \overline{\mathbf{Y}}\_{k}^{\left(\mathcal{I}\_{t}\right)} ; \mathbf{w}^{(t)})$ for $k \in[K]$. Is it not that the privacy for the server should be proved in terms of the decoded gradient? In other words, Theoren 2 should be stated in terms of $\widetilde{\boldsymbol{g}}(\overline{\mathbf{X}}\_{j}^{(\mathcal{I}\_{t})}, \overline{\mathbf{Y}}\_{j}^{\left(\mathcal{I}\_{t}\right)} ; \mathbf{w}^{(t)})$? Overall, it is not clear how the server only obtains an aggregation of gradients. It is important to expand on this.
>
> $\textbf{Answer 2:}$ Thank you for this comment. Our proposed framework can aggregate the local computation results from the clients while providing a privacy guarantee such that the server learns no information about the private datasets of clients from a single local computation result beyond the model parameters.
>
> (1) Aggregation of gradients: As discussed in $\textbf{Answer 1.2}$ (In $\textbf{Part 3}$), the server obtains the stochastic gradients $\widetilde{\boldsymbol{g}}(\overline{\mathcal{D}}^{(\mathcal{I}\_{t})}\_{k};\mathbf{w}^{(t)})$ for $k \in [K]$ after decoding. Every $\widetilde{\boldsymbol{g}}(\overline{\mathcal{D}}^{(\mathcal{I}\_{t})}\_{k};\mathbf{w}^{(t)})$ is an aggregation of gradients over the global mini-batch $\overline{\mathcal{D}}^{(\mathcal{I}\_{t})}\_{k}$.
>
> (2) Privacy guarantee: As discussed in Appendix B, since the mutual information $I(\overline{\mathbf{X}}^{(i)},\overline{\mathbf{Y}}^{(i)};\widetilde{\boldsymbol{g}}(\widetilde{\mathbf{X}}\_{j}^{(\mathcal{I}\_{t})},\widetilde{\mathbf{Y}}\_{j}^{(\mathcal{I}\_{t})};\mathbf{w}^{(t)})|\mathbf{w}^{(t)})$ equals to zero for any $i,j \in [N]$, the server learns no information about the private datasets of clients from a single local computation result beyond the model parameters. This property, which is desired in secure aggregation, helps the local update against the model inversion attacks.
>
> Nevertheless, the privacy leakage problem still exists as the model parameters $\mathbf{w}^{(t)}$ and the aggregated results (e.g., $\widetilde{\boldsymbol{g}}(\overline{\mathcal{D}}\_{k};\mathbf{w}^{(t)})$) contain information about the datasets, i.e., the mutual information $I(\overline{\mathcal{D}};\mathbf{w}^{(t)}) \neq 0$ and the conditional mutual information $  I(\overline{\mathcal{D}};\widetilde{\boldsymbol{g}}(\overline{\mathcal{D}}\_{k};\mathbf{w}^{(t)})|\mathbf{w}^{(t)}) \neq 0$ for $i \in [N]$ and $k \in [K]$. Our proposed method, as well as the existing secure aggregation approaches, cannot achieve "perfect privacy", which is highly challenging and is an active area of research.
>
> > $\textbf{Question 3:}$ In FL all communication typically happens through the server. It is important to mention how a client will securely communicate the secret shared data to all other clients.
>
> $\textbf{Answer 3:}$ We thank the reviewer for this comment. From our perspective, how to construct secure communications among clients is orthogonal to the "secret data sharing" in our method. For example, public-key cryptography can be used for encrypting communication to provide confidentiality, where the server can be a relay node and cannot access the messages.
>
> > $\textbf{Question 4.1:}$ The proposed scheme involves several approximations. It seems like these approximations will negatively impact the accuracy. However, performance evaluation in Fig. 2 depicts that the proposed method consistently outperforms FedAvg for all datasets. It would be important to discuss why the approximations do not impact accuracy.
>
> $\textbf{Answer 4.1:}$ We would like to clarify two points:
>
> (1) These approximations do degrade the accuracy of the proposed method in the classification task. In the experiments, we select the centralized training scheme as a baseline, which uses ReLU as the activation function, adopts cross-entropy as the loss function, and trains the model over the real field. Ideally, without any approximation, our method could approach the performance of this centralized setting. As reported in Table 2, however, our proposed method achieves lower accuracy than this centralized training scheme. This is attributed to these approximations.
>
> (2) The reason why our proposed method can consistently perform better than FedAvg is that FedAvg suffers from the non-IID dataset and client dropouts, while our method can effectively alleviate these two problems. Although several approximations degrade the performance of the proposed work, our method still outperforms all the federated learning baselines.

---

> ### Author Response · Authors · 2022-08-02
> **Response to Reviewer ULSx (Part 3)**
>
> > The following is the response to $\textbf{Question 1}$ in $\textbf{Part 2}$
>
> $\textbf{Answer 1.2:}$ We would like to emphasize that the server performs $\textbf{joint decoding}$ (rather than separate decoding) in our method. More details are as follows:
>
> Table 3: Primary notations and descriptions
>
> |Notation|Description|
> | :--: | :--: |
> |$(\overline{\mathbf{X}}^{(i)},\overline{\mathbf{Y}}^{(i)})$| Transformed dataset at client $i$  over finite filed $\mathbb{F}_{p}$ |
> |$\overline{\mathcal{D}}$| Concatenation of datasets $(\overline{\mathbf{X}}^{(i)},\overline{\mathbf{Y}}^{(i)})$ for $i \in [N]$ |
> | $(\overline{\mathbf{X}}\_{k}^{(i)},\overline{\mathbf{Y}}\_{k}^{(i)})$ |$k$-th data shard at client $i$|
> | $(\widetilde{\mathbf{X}}\_{j}^{(i)},\widetilde{\mathbf{Y}}\_{j}^{(i)})$ |Encoded dataset sent from client $i$ to client $j$|
> | $(\widetilde{\mathbf{X}}\_{j},\widetilde{\mathbf{Y}}\_{j})$| Concatenation of all the received encoded datasets at client $j$ |
> |$\mathbf{C}^{(t)}$|row selection matrix for data sampling in round $t$|
> | $(\widetilde{\mathbf{X}}\_{j}^{\left(\mathcal{I}\_{t}\right)}, \widetilde{\mathbf{Y}}\_{j}^{\left(\mathcal{I}\_{t}\right)})$ | mini-batch sampled from $(\widetilde{\mathbf{X}}\_{j},\widetilde{\mathbf{Y}}\_{j})$  at client $j$ in round $t$ |
> | $\overline{\mathcal{D}}\_{k}^{\left(\mathcal{I}\_{t}\right)} \triangleq  (\overline{\mathbf{X}}\_{k}^{\left(\mathcal{I}\_{t}\right)},\overline{\mathbf{Y}}\_{k}^{\left(\mathcal{I}\_{t}\right)})$ | $k$-th global mini-batch sampled from $\overline{\mathcal{D}}$ in round $t$ |
> |$\mathbf{u}\_{\mathcal{I}\_{t}}(z),\mathbf{v}\_{\mathcal{I}\_{t}}(z)$| $(K+T-1)$-order encoding polynomials |
>
> In each round, all the clients use the same row selection matrix $\mathbf{C}^{(t)}$ for data sampling. The mini-batch $(\widetilde{\mathbf{X}}\_{j}^{\left(\mathcal{I}\_{t}\right)}, \widetilde{\mathbf{Y}}\_{j}^{\left(\mathcal{I}\_{t}\right)})$ is sampled from the encoded dataset $(\widetilde{\mathbf{X}}\_{j},\widetilde{\mathbf{Y}}\_{j})$ at client $j$. Meanwhile, $K$ $\textbf{global mini-batches}$ $\overline{\mathcal{D}}^{(\mathcal{I}\_{t})}\_{k} \triangleq (\overline{\mathbf{X}}\_{k}^{(\mathcal{I}\_{t})},\overline{\mathbf{Y}}\_{k}^{(\mathcal{I}\_{t})})$ for $k \in [K]$ are selected from the global dataset $\overline{\mathcal{D}}$ by $\overline{\mathbf{X}}\_{k}^{(\mathcal{I}\_{t})} =  \mathbf{C}^{(t)} [\overline{\mathbf{X}}\_{k}^{(1)T},\ldots,\overline{\mathbf{X}}\_{k}^{(N)T}]^{T}$ and $\overline{\mathbf{Y}}\_{k}^{(\mathcal{I}\_{t})} =   \mathbf{C}^{(t)} [\overline{\mathbf{Y}}\_{k}^{(1)T},\ldots,\overline{\mathbf{Y}}\_{k}^{(N)T}]^{T}$. (Note that $\overline{\mathcal{D}}^{(\mathcal{I}\_{t})}\_{k} \triangleq (\overline{\mathbf{X}}\_{k}^{(\mathcal{I}\_{t})},\overline{\mathbf{Y}}\_{k}^{(\mathcal{I}\_{t})})$ is $\textbf{not}$ a mini-batch of the local dataset $(\overline{\mathbf{X}}^{(k)},\overline{\mathbf{Y}}^{(k)})$ at client $k$.)
>
> The server decodes $K$ stochastic gradients $\widetilde{\boldsymbol{g}}(\overline{\mathcal{D}}^{(\mathcal{I}\_{t})}\_{k};\mathbf{w}^{(t)}) \triangleq \widetilde{\boldsymbol{g}}(\overline{\mathbf{X}}\_{k}^{(\mathcal{I}\_{t})},\overline{\mathbf{Y}}\_{k}^{(\mathcal{I}\_{t})};\mathbf{w}^{(t)})$ for $k \in [K]$ after receiving at least $ \text{deg} (\boldsymbol{g}) (K+T-1)+1 $ local computation results $\widetilde{\boldsymbol{g}}(\widetilde{\mathbf{X}}\_{j}^{(\mathcal{I}\_{t})},\widetilde{\mathbf{Y}}\_{j}^{(\mathcal{I}\_{t})};\mathbf{w}^{(t)})$ for $j \in [N]$. Each result $\widetilde{\boldsymbol{g}}(\widetilde{\mathbf{X}}\_{j}^{(\mathcal{I}\_{t})},\widetilde{\mathbf{Y}}\_{j}^{(\mathcal{I}\_{t})};\mathbf{w}^{(t)}) = \widetilde{\boldsymbol{g}}(\mathbf{u}\_{\mathcal{I}\_{t}}(\alpha_{j}),\mathbf{v}\_{\mathcal{I}\_{t}}(\alpha_{j});\mathbf{w}^{(t)})$ amounts to an evaluation point at $z = \alpha\_{j} \in \mathbb{F}\_{p}$ of the composed polynomial  $\widetilde{\boldsymbol{g}}(\mathbf{u}\_{\mathcal{I}\_{t}}(z),\mathbf{v}\_{\mathcal{I}\_{t}}(z);\mathbf{w}^{(t)})$. As the degree of this polynomial is $ \operatorname{deg} (\boldsymbol{g}) (K+T-1)$, the server needs at least $ \operatorname{deg} (\boldsymbol{g}) (K+T-1)+1 $ evaluation points to reconstruct it. After recovering the coefficients of the polynomial $\widetilde{\boldsymbol{g}}(\mathbf{u}\_{\mathcal{I}\_{t}}(z),\mathbf{v}\_{\mathcal{I}\_{t}}(z);\mathbf{w}^{(t)})$, the server decodes the gradients $\widetilde{\boldsymbol{g}}(\overline{\mathcal{D}}^{(\mathcal{I}\_{t})}\_{k};\mathbf{w}^{(t)}) \triangleq  \widetilde{\boldsymbol{g}}(\overline{\mathbf{X}}\_{k}^{\left(\mathcal{I}\_{t}\right)}, \overline{\mathbf{Y}}\_{k}^{\left(\mathcal{I}\_{t}\right)};\mathbf{w}^{(t)}) = \widetilde{\boldsymbol{g}}(\mathbf{u}\_{\mathcal{I}\_{t}}(\beta_{k}),\mathbf{v}\_{\mathcal{I}\_{t}}(\beta_{k});\mathbf{w}^{(t)})$  by letting $z=\beta_{k}$ for $k \in [K]$. Finally, the decoded gradients are averaged after being converted from the finite field to the real field.

---

> ### Author Response · Authors · 2022-08-02
> **Response to Reviewer ULSx (Part 2)**
>
> > The following content is part of the response to $\textbf{Weakness 1}$ in $\textbf{Part 1}$
>
> Table 2: Communication complexity comparison
>
>  || Data sharing (before training starts) | Coded masks sharing among clients ($\tau$ training rounds) | Local model uploading ($\tau$ training rounds) |    Coded masks uploading ($\tau$ training rounds)    | Global model downloading ($\tau$ training rounds) |
>  | :---------------------: | :-----------------------------------: | :--------------------------------------------------------: | :--------------------------------------------: | :--------------------------------------------------: | :-----------------------------------------------: |
>  |FedAvg|$-$|$-$|$\mathcal{O}(\tau  d_{w})$           |                         $-$                          |            $\mathcal{O}(\tau N d_{w})$            |
>  | FedAvg with LightSecAgg |                  $-$                  |  $\mathcal{O}\left( \frac{\tau N^{2}d_{w}}{R-T} \right)$   |           $\mathcal{O}(\tau  d_{w})$           | $\mathcal{O}\left( \frac{\tau d_{w} R}{R-T}\right) $ |            $\mathcal{O}(\tau N d_{w})$            |
>  |DReS-FL|$\mathcal{O}(N^{2}/K)$         |$-$  |           $\mathcal{O}(\tau  d_{w})$           |                         $-$                          |            $\mathcal{O}(\tau N d_{w})$            |
>
> These two tables show that the computational and communication complexities of both the secure aggregation method and our DReS-FL method increase quadratically with the number of clients $N$. Nevertheless, the main drawback of the secure aggregation methods is that clients need to generate new masks in each round, and their computational and communication complexities increase linearly with the number of training rounds. In comparison, the computational and communication costs introduced by the data sharing in our method are one-time costs, which are independent of the number of training rounds. Particularly, when the number of training rounds is very large, the proposed DReS-FL method could achieve lower computational and communication complexities compared with the secure aggregation protocols.
>
> Overall, as the extra costs introduced by the data sharing phase do not increase with the training round, the authors believe the complexities of our method are acceptable.
>
> > $\textbf{Weakness 2:}$ The paper tackles the client dropouts problem. However, it looks like all clients need to be present during the phase of secret sharing data. This assumption needs to be clearly stated and it would be good to add a discussion around this. In particular, if the communication cost of transferring secret shared data to other clients is large, then this phase will take considerable time. In this case, not allowing any dropouts during secret sharing can be a major limitation.
>
> $\textbf{Response 2:}$ This is an interesting question. Our proposed method aims at solving the dropout problem during the training process. Generally, federated training has an initialization phase before the training starts to confirm the participation of clients, in which period clients can share coded data. If some clients are unavailable during this phase, the server can simply remove these clients from the federated learning system. Alternatively, the server can wait for a while to ensure a sufficient number of clients have completed the data sharing step. Moreover, as discussed in $\textbf{Response 1}$, the communication overhead in data sharing is tolerable.
>
> > $\textbf{Question 1:}$ The paper does not give details on the gradient computation of the secret shared data. It simply mentions that the gradient (when the loss is MSE and non-linearity is a square function) would be a multivariate polynomial in $(\overline{\mathbf{X}}^{(i)}, \overline{\mathbf{Y}}^{(i)})$ and parameters w. Without any details, it is difficult to understand how the server decodes $\widetilde{\boldsymbol{g}}(\widetilde{\mathbf{X}}\_{j}^{(\mathcal{I}\_{t})}, \widetilde{\mathbf{Y}}\_{j}^{\left(\mathcal{I}\_{t}\right)} ; \mathbf{w}^{(t)})$ in shard separately. It is important to add details and provide a proof of how the decoding happens. Currently, it is not even clear how the gradient $\widetilde{\boldsymbol{g}}(\widetilde{\mathbf{X}}\_{j}^{(\mathcal{I}\_{t})}, \widetilde{\mathbf{Y}}\_{j}^{\left(\mathcal{I}\_{t}\right)} ; \mathbf{w}^{(t)})$ is defined. Is this an average of the gradients in batch $\mathcal{I}\_{t}$?
>
> $\textbf{Answer 1.1:}$ The local gradient computation step in the proposed DReS-FL method is roughly the same as that in the standard FL framework. The only differences are that we use the encoded samples $(\widetilde{\mathbf{X}}\_{j}^{(\mathcal{I}\_{t})}, \widetilde{\mathbf{Y}}\_{j}^{\left(\mathcal{I}\_{t}\right)})$ as the inputs of the gradient function and the computation is over a finite field. Due to the character constraint, please refer to $\textbf{Answer1.2}$ in $\textbf{Part 3}$ for more details.

---

> ### Author Response · Authors · 2022-08-02
> **Response to Reviewer ULSx (Part 1)**
>
> __We are grateful to the reviewer for the effort and time spent on our manuscript. We would like to clarify/correct a few key points in the reviewer's comments, with the hope that this will help highlight the key contributions of our work. The detailed responses to your comments on the weaknesses and questions are itemized in the following.__
>
> > $\textbf{Weakness 1:}$ The proposed solution requires each client to secretly share their data with all other clients. Clearly, this is going to incur a substantial burden in terms of computation and communication costs. The paper does not explicitly characterize the computation and communication costs for secretly sharing client data. (It just mentions that the "sharing" parameter impacts the costs, but the explicit analysis seems to be missing.)
>
> $\textbf{Response 1:}$ Thank you for this comment. Please note that we had provided a complexity analysis in Section C of Appendix, which demonstrates that our proposed method achieves comparable or even lower communication and computational costs than the secure aggregation methods. In the following, we specify the complexities of the proposed DRes-FL method, the secure sharing methods, as well as the FedAvg method for comparison.
>
> We analyze the communication and computational complexities of the proposed DReS-FL framework with respect to the parameters $(N,T,K,\tau,d_{w},b_{g})$. Parameter $N$ is the number of clients, and $T$ is a privacy parameter in Lagrange coding [1]. Parameter $K$ denotes the number of shards in the local datasets. A large value of $K$ reduces the communication overhead in secret data sharing and the local computation loads of clients. In federated training, the parameter $\tau$ corresponds to the number of training rounds. Parameters $d_{w}$ and $b_{g}$ denote the model size and the global batch size, respectively. Before training starts, each client's computation cost for Lagrange coding and communication complexity for data sharing are $\mathcal{O}(N \log ^{2}(K+T) \log \log (K+T))$ and $\mathcal{O}(N / K)$, respectively [1]. In each round of federated training, the local computation complexity is $\mathcal{O}(d_{w}b_{g}/K)$, and the model uploading cost is $\mathcal{O}(d_{w})$. Besides, the communication overhead of the server for model distributing is $\mathcal{O}(Nd_{w})$, and the model decoding complexity by polynomial interpolation is $\mathcal{O}(R\log^{2}R \log \log R  d_{w})$, where $R$ denotes the minimum uploads needed for gradient decoding.
>
> Different from our method, secure aggregation approaches (e.g., LightSecAgg [2]) generate random masks to protect the local model parameters. In each round, clients first share coded masks with each other, which allows for aggregating the masked models at the server. As some clients may drop out of the training process unexpectedly, the surviving clients upload the shared information belonging to the dropped clients to reconstruct the aggregated model. The complexity comparisons among FedAvg, FedAvg with LightSecAgg [2], and DReS-FL are summarized in the following tables:
>
>  Table 1: Computational complexity comparison
>
>  |                         |    Lagrangian coding (before training starts)     |    Generating coded random masks ($\tau$ training rounds)    |  Local model update ($\tau$ training rounds)   |     Global model aggregation ($\tau$ training rounds)      |
>  | :---------------------: | :-----------------------------------------------: | :----------------------------------------------------------: | :--------------------------------------------: | :--------------------------------------------------------: |
> |         FedAvg          |                        $-$                        |                             $-$                              | $\mathcal{O}\left(\tau d_{w} b_{g} / N\right)$ |                 $\mathcal{O}(\tau Nd_{w})$                 |
>  | FedAvg with LightSecAgg |                        $-$                        | $\mathcal{O}\left(\frac{\tau d_{w} N^{2} \log N}{R-T}\right)$ | $\mathcal{O}\left(\tau d_{w} b_{g} / N\right)$ | $\mathcal{O}\left(\frac{\tau d_{w} R \log R}{R-T} \right)$ |
>  |         DReS-FL         | $\mathcal{O}(N^{2}\log ^{2}(K+T)\log \log (K+T))$ |                             $-$                              | $\mathcal{O}\left(\tau d_{w} b_{g} / N\right)$ |     $\mathcal{O}(\tau d_{w}R \log ^{2}R\log \log R )$      |
>
>  [1] Yu et al. "Lagrange coded computing: Optimal design for resiliency, security, and privacy," AISTATS, 2019.
>
>  [2] So et al. "Lightsecagg: a lightweight and versatile design for secure aggregation in federated learning," MLSys, 2022.
>
> ** **
> __Due to the character constraint, more discussion is deferred to $\textbf{Part 2}$ below.__

---

> ### Author Response · Authors · 2022-08-08
> **Please can you respond to our rebuttal?**
>
> Dear __Reviewer ULSx__,
>
> We are approaching the end of the discussion phase, and we have unfortunately received no feedback from you on our rebuttal.
>
> Please can we kindly ask you to take a look at our response (especially __Part 3__) and let us know whether we addressed your concerns? As the reviewer _appeared to misunderstand_ the gradient decoding process in the proposed DReS-FL framework, we have provided more details about it in __Part 3__.
>
> Thank you very much again for the time you spent reviewing.
>
> Paper6370 Authors

---

> > ### Comment · Reviewer_ULSx · 2022-08-09
> > **Thanks for the detailed comments. There are still a few concerns.**
> >
> > Dear Authors,
> >
> > Thanks for the detailed answers to my comments. This addresses several of my concerns. I still have a few concerns.
> >
> > 1. I understand the decoding process at the server now. However, it would be great to add more details on how clients compute gradients (as mentioned towards the end of Question 1).
> > * Towards this end, the example discussed in Answer 10 is helpful. It would be great to add a vector version of this example in the appendix, and discuss how degree of $g$ is a function of the number of layers.
> > * Can the proposed method handle convolutional layers, max pooling layers, dropouts, batch norm, skip connections?
> > * In their answer, the authors mention that for a neural network with $L$ layers, the degree of $g$ is $2^{L+1}$. For practical neural networks, which are often deep, this would result in a very large degree requiring large $N$ or making $K$, $T$, $D$ very small. Can the authors elaborate on this?
> >
> > 2. Can the authors give some more details on how the communication cost is $O(N^2/K)$? Would it not be $(N^2/K \times (d_x + d_y))$? In other words, communication cost grows with the number of features. For VGG19, $d_x$ would be number of features from the last conv layer, which is quite large.
> >
> > 3. In Answer 3, the authors mention that 'how to construct secure communications is orthogonal to the proposed method'. However, such methods, e.g., public-key encryption on top of secret sharing, will impact the actual communication cost, which may make the scheme much less practical.
> >
> > 4. In Appendix B.1, how is $I(\bar{X}^{(i)},\bar{Y}^{(i)}; \tilde{X}^{(I_t)},\tilde{Y}^{(I_t)}, w^t | w^t) = 0$? Is it assumed that data samples of different clients are independent of each other? It is important to discuss any assumptions.
> >
> > 5. The authors mention that their scheme is focused on cross-device setting, which may consists of tens of thousands to millions of devices, see e.g., [Bonawitz et al., SysML 2019]. Orchestrating secret sharing for such a large number of devices at the setup phase may be limiting.
> >
> > 6. It is interesting that even after using MSE instead of Cross-Entropy loss, squared non-linearity instead of ReLU, and quantization to integers, the loss in accuracy is less that 1-4% even for complex datasets like CIFAR-10/CIFAR-100. Is it because a pre-trained network is used and only last few fully connected layers are fine tuned?
> >
> > 7. In Answer 5, the loss $\ell(\mathbf{w})$ is the population risk, right? The empirical risk would be $\frac{1}{m}\sum_{i = 1}^{m}|Y - f(X_i,w)|^2$, where $m$ is the number of samples (unless I am missing something).
> >
> > 8. Once a neural network is converted as a multivariate polynomial function, the proposed scheme can be considered as a direct application of Lagrange coded computing (LCC). Can the authors highlight the novelty of the proposed method over LCC?

---

> > > ### Comment · Reviewer_ULSx · 2022-08-09
> > > **Secret-sharing communication cost for experimental setup**
> > >
> > > The authors clarify the communication cost in Answer 2 to Reviewer kTxQ. I am sorry that I missed it earlier. It is mentioned to be proportional to $\frac{\sum_{i=1}^{N}m_i(d_x+d_y)(N-1)N}{K}$.
> > >
> > > If we consider this for CIFAR-100 experimental setup, we get the following numbers. (Please correct me if I am making any mistake.)
> > >
> > > Consider experimental setup for CIFAR-100 with 50000 training examples divided into $N=20$ shards. Then, $m_i = 50000/20 = 2500$, $d_y =100$, $d_x \approx 25088$ for the output of conv layer of VGG19, and $K = 1$ as considered in Sec. 6.1. This yields that the communication cost of secret sharing is proportional to $\mathbf{4.78\times 10^{11}}$ scalars, each of which lies in $\mathbb{F}_p$, where $p = 10^{71} + 273$ (Table 2 in Appendix). The cost will even increase when public-key encryption is used to establish secure communications! Considering this, it seems like it would be more appropriate if costs are measured in terns of actual payload size.

---

> > > > ### Author Response · Authors · 2022-08-09
> > > > **Communication complexity (Second Response: Part 3)**
> > > >
> > > > In this part, the authors would like to discuss the communication complexity of the proposed method and address the concerns in $\textbf{Comment 2}$, $ \textbf{Comment 3}$, $\textbf{Comment 5}$ and the above post.
> > > >
> > > > But first, please allow the authors to recap the main contributions of our work. Our DReS-FL framework aims at tackling two critical problems of FL, namely, (1) __non-IID data distribution__ and (2) __client dropouts__ while (3) providing the same __security guarantee__ as the _secure aggregation_ approaches. The key idea is to utilize Lagrange coding to secretly share the private datasets among clients so that the effects of non-IID distribution and client dropouts can be compensated during local gradient computations. Compared with the standard federated learning methods (e.g. FedAvg), our method achieves better model performance and provides an extra security guarantee on local computation results.
> > > >
> > > > However, our method leads to a relatively high communication overhead caused by secret data sharing. Similar to the existing secure aggregation approaches, the communication complexity of our method increases quadratically with the number of clients. But, our method has one main difference: The data sharing phase happens before the training starts, which means that the extra communication overhead is a __one-time cost__ and is independent of the training round. In comparison, the secure aggregation approaches need to generate and share new masks to maintain the security guarantee in each round, which incurs high communication overhead when the number of training rounds is large. Therefore, our method achieves comparable or even lower communication costs than the secure aggregation approaches. The authors believe that the extra costs caused by our method are acceptable and tolerable. How to effectively reduce the communication complexity while providing a security guarantee is still highly challenging and is an active area of research.
> > > >
> > > > The authors sincerely hope the reviewer will be satisfied with the response.
> > > >
> > > > > In the following, we discuss the example provided by the reviewer.
> > > >
> > > > Please note that the prime number $p = 10^{71}+273$ in our work is a feasible value for model training, but not an optimal prime number that uses the smallest bits to represent each scalar in the gradient. Previous work [1] on privacy-preserving learning shows that the finite field with the prime number $p=2^{26}-5$  is sufficient for the CIFAR-10 training. Thus, our work has the potential to further reduce the field size, but this is beyond the scope of this paper.
> > > >
> > > > In the CIFAR-100 experiment, each entry in the dataset is quantized to 7 bits (since we need at least 7 bits to represent 100 classes), and the dimension $d_y = 1$. In the data sharing phase, each client needs to transmit $2.4 \times 10^{10}$ scalars, corresponding to 20GB. Assume the clients are supported by 5G (rate: 50 Mbps), the communication latency is around 3000 seconds for data sharing. When the clients are supported by 5G with a transmission rate of 1 Gbps, the data sharing phase tasks less than 3 minutes.
> > > >
> > > > In comparison, secure aggregation protocols need to share new masks of the model with other clients in each round, which is proportional to the number of parameters $d_{w}$ in the neural network. Take the LightSecAgg method [2] as an example. The local model is defined over a finite field $\mathbb{F}_{q}$, where $q = 2^{32}-5$. The communication cost of coded mask sharing of each client is proportional to 8.5GB, which corresponds to 1400 seconds (over a 50 Mbps 5G connection) or 70 seconds (over a 1 Gbps 5G connection).
> > > >
> > > > After several training rounds, the secure aggregation protocols may lead to higher communication costs than our method.
> > > >
> > > > [1] So et al. "A scalable approach for privacy-preserving collaborative machine learning," NeurIPS, 2020
> > > >
> > > > [2] So et al. "Lightsecagg: a lightweight and versatile design for secure aggregation in federated learning," MLSys, 2022.

---

> > > > > ### Comment · Reviewer_ULSx · 2022-08-09
> > > > > **Thank you carefully considering the comments**
> > > > >
> > > > > Thank you to the authors for carefully considering the concerns and providing detailed answers. Hope that these discussions will help the authors to improve the presentation.
> > > > >
> > > > > After the discussions, I believe that the paper is technically sound. However, the practicality of the proposed scheme seems quite limited due to huge communication cost. (As the authors mention, over a 50 Mbps 5G connection, it would take >50 minutes for the setup phase, which may be infeasible for cross-device setting with limited device availability.) Also, when using pre-trained models, each client needs to first compute logits/embeddings for their entire dataset before secret sharing happens. This computation time will add to the setup time. Considering these points, I increase my score from 3 to 4.

---

> > > > > > ### Author Response · Authors · 2022-08-10
> > > > > > **Thank you for your prompt reply (Third Response)**
> > > > > >
> > > > > > __We are glad to see that the reviewer has raised the evaluation score. Based on the response above, the authors would like to take the last chance to clarify two points.__
> > > > > >
> > > > > > > $\textbf{Comment 1:}$ The practicality of the proposed scheme seems quite limited due to the huge communication cost. (As the authors mention, over a 50 Mbps 5G connection, it would take >50 minutes for the setup phase, which may be infeasible for cross-device settings with limited device availability.)
> > > > > >
> > > > > > $\textbf{Point 1:}$ In the previous response, the authors assume the communication rate for data sharing is 50 Mbps, which is roughly the lowest rate of 5G. As mentioned in IMT-2020 Standardization, 5G is designed to deliver data rates up to 20 Gbps. When the clients are supported by 5G with a transmission rate of 1 Gbps, the data sharing phase tasks less than 3 minutes. The authors believe this cost is acceptable, and the communication complexity is not necessarily a bottleneck, thanks to the advanced wireless communication techniques.
> > > > > >
> > > > > > The main focuses of our work are on (1) data heterogeneity, (2) client dropouts, and (3) security guarantee. Improving the federated training efficiency is an interesting direction for our future research.
> > > > > >
> > > > > >
> > > > > > > $\textbf{Comment 2:}$ When using pre-trained models, each client needs to first compute logits/embeddings for their entire dataset before secret sharing happens. This computation time will add to the setup time.
> > > > > >
> > > > > > $\textbf{Point 2:}$ In the experiments (on CIFAR-10, CIFAR-100, and SVHN datasets), both our method and the baseline methods adopt pre-trained models for feature extraction. This means that our method does not lead to extra computation costs.

---

> > > ### Author Response · Authors · 2022-08-09
> > > **Thank you for your reply (Second Response: Part 2)**
> > >
> > > > $\textbf{Comment 4:}$ In Appendix B.1, how is $I\left(\overline{\mathbf{X}}^{(i)}, \overline{\mathbf{Y}}^{(i)} ; \widetilde{\mathbf{X}}\_{j}^{\left(\mathcal{I}\_{t}\right)}, \widetilde{\mathbf{Y}}\_{j}^{\left(\mathcal{I}\_{t}\right)}, \mathbf{w}^{(t)} \mid \mathbf{w}^{(t)}\right) = 0$? Is it assumed that data samples of different clients are independent of each other? It is important to discuss any assumptions.
> > >
> > > $\textbf{Response 4:}$ Note that  $(\widetilde{\mathbf{X}}\_{j}^{\left(\mathcal{I}\_{t}\right)}, \widetilde{\mathbf{Y}}\_{j}^{\left(\mathcal{I}\_{t}\right)})$ is a mini-batch of $(\widetilde{\mathbf{X}}\_{j}, \widetilde{\mathbf{Y}}\_{j})$, and $(\overline{\mathbf{X}}^{(i)}, \overline{\mathbf{Y}}^{(i)})$ is a subset of $\overline{\mathcal{D}}$. As we have mentioned in Theorem 1 that $I(\overline{\mathcal{D}};\widetilde{\mathbf{X}}\_{j},\widetilde{\mathbf{Y}}\_{j}) = 0 $,  we get $I\left(\overline{\mathbf{X}}^{(i)}, \overline{\mathbf{Y}}^{(i)} ; \widetilde{\mathbf{X}}\_{j}^{\left(\mathcal{I}\_{t}\right)}, \widetilde{\mathbf{Y}}\_{j}^{\left(\mathcal{I}\_{t}\right)}, \mathbf{w}^{(t)} \mid \mathbf{w}^{(t)}\right) = 0$.
> > >
> > > > The response to $\textbf{Comment 5}$ has been deferred to __Second Response: Part 3__
> > >
> > > > $\textbf{Comment 6:}$ It is interesting that even after using MSE instead of Cross-Entropy loss, squared non-linearity instead of ReLU, and quantization to integers, the loss in accuracy is less that 1-4% even for complex datasets like CIFAR-10/CIFAR-100. Is it because a pre-trained network is used and only the last few fully connected layers are fine tuned?
> > >
> > > $\textbf{Response 6:}$ The authors believe that the pre-trained model helps a lot to improve the accuracy.
> > >
> > > > $\textbf{Comment 7:}$ In Answer 5, the loss $ℓ(w)$ is the population risk, right? The empirical risk would be $\frac{1}{m}∑_{i=1}^{m}|Y−f(X_i,w)|^{2}$, where m is the number of samples (unless I am missing something).
> > >
> > > $\textbf{Response 7}$ In our convergence analysis, we consider the empirical risk with the loss function $\ell(\mathbf{w}) \triangleq \mathbb{E}\_{(\overline{\mathbf{X}},\overline{\mathbf{Y}}) \sim \overline{\mathcal{D}}} \left[ \|\overline{\mathbf{Y}}-\boldsymbol{f}(\overline{\mathbf{X}} ; \mathbf{w})\|\_{2}^{2} \right]$. Here $(\overline{\mathbf{X}},\overline{\mathbf{Y}}) \sim \overline{\mathcal{D}}$ means a data sample is randomly drawn from the $\textbf{training}$ dataset $\overline{\mathcal{D}}$. In other words, the loss function is defined on all samples of the training dataset $\overline{\mathcal{D}}$. For an explicit expression, we introduce notations $(\bar{x}\_{j}, \bar{y}\_{j})$ as the $j$-th training data sample in the training dataset $\overline{\mathcal{D}}$. Thus, the loss function can be rewritten as $\mathbb{E}\_{(\overline{\mathbf{X}},\overline{\mathbf{Y}}) \sim \overline{\mathcal{D}}} \left[ \|\overline{\mathbf{Y}}-\boldsymbol{f}(\overline{\mathbf{X}} ; \mathbf{w})\|\_{2}^{2} \right] \equiv \frac{1}{m} \sum_{j=1}^{m} \|{\bar{y}}\_{j}-\boldsymbol{f}(\bar{x}\_{j} ; \mathbf{w})\|\_{2}^{2}$, which is exactly the empirical risk. We hope this explanation is more clear.
> > >
> > > > $\textbf{Comment 8:}$ Once a neural network is converted as a multivariate polynomial function, the proposed scheme can be considered as a direct application of Lagrange coded computing (LCC). Can the authors highlight the novelty of the proposed method over LCC?
> > >
> > > $\textbf{Response 8:}$ Thank you for this advice. We will emphasize the importance of the LCC to our work in the final version of the manuscript.

---

> > > ### Author Response · Authors · 2022-08-09
> > > **Thank you for your reply (Second Response: Part 1)**
> > >
> > > __We are grateful to see that some concerns from the reviewer have been addressed__. For the new comments raised by the reviewer, we have provided point-to-point responses in the following.
> > >
> > > > $\textbf{Comment 1:}$ I understand the decoding process at the server now. However, it would be great to add more details on how clients compute gradients (as mentioned towards the end of Question 1).
> > > >
> > > > - Towards this end, the example discussed in Answer 10 is helpful. It would be great to add a vector version of this example in the appendix and discuss how the degree of gradient function is a function of the number of layers.
> > > > - Can the proposed method handle convolutional layers, max-pooling layers, dropouts, batch norm, and skip connections?
> > > > - In their answer, the authors mention that for a neural network with L layers, the degree of g is 2L+1. For practical neural networks, which are often deep, this would result in a large degree requiring large N or making K, T, and D very small. Can the authors elaborate on this?
> > >
> > > $\textbf{Response 1:}$ Thank you for this comment.
> > >
> > > - We are happy to see that our example in __Answer 10__ is helpful and the reviewer understands the decoding process. Following the suggestion from the reviewer, we will add more details about the gradient computation and provide more discussion about how to compute the degree of the gradient in the final version of the paper.
> > > - Our method can handle convolutional layers and skip connections. However, max-pooling layers, dropout layers, and batch normalization layers can not be involved in the proposed polynomial integer neural network (PINN). This is because our method requires the gradient to be a polynomial function. But, our method can utilize the pre-trained models for feature extraction, which do not have the constraint on the type of layer.
> > > - Increasing the number of devices $N$ lets the devices cooperatively train a deep model. Besides, we can select a pre-trained model for feature extraction and fine-tune a shallow model during the federated training. This solution can largely reduce the degree of the gradient.
> > >
> > > > The responses to $\textbf{Comment 2}$ and $\textbf{Comment 3}$ have been deferred to __Second Response: Part 3__

---

### Author Response · Authors · 2022-08-07
**General Response**

We are very thankful for the area chair to coordinate the review of our manuscript and grateful to the reviews for their valuable feedback. We very much appreciate the assessment of work as __novel (Reviewer FXUt, gqjh)__, __interesting (Reviewer FXUt, kTxQ)__, and __important (Review ULSx, kTxQ)__ with __theoretical analyses (Reviewer FXUt, gqjh, kTxQ)__.

The authors have provided point-to-point responses to the comments raised by the reviewers. A summary of them is as follows:

1. To address __Reviewer ULSx__ and __Reviewer kTxQ__'s concern about the __complexity__ of the proposed method, we have discussed that the extra costs introduced by the data sharing phase in our method are __one-time costs__, which are independent of the number of training rounds. As the complexity of the federated training gradually increases with the training round, the extra costs in our method become less and less important. Compared with the secure aggregation approaches that introduce extra overheads in each round caused by mask sharing, our method __achieves comparable or even lower communication/computation costs__ while (1)  providing the same security guarantee and (2) further tackling the data heterogeneity and dropout problems in FL.  Moreover, as discussed in the __Third Response__ to __Reviewer ULSx__, the costs in the data sharing phase do not limit the utility of the proposed method.
2. To answer the question from __Reviewer ULSx__ about how to decode the gradient during the training process, we have described the technical details in __Part 2__ and __Part 3__ of the response. Please note that the server adopts polynomial interpolation [1] to perform __joint decoding__ rather than __separate decoding__.
3. As suggested by __Reviewer gqjh__, we have added more details about the experimental setups.
4. Following the advice from __Reviewer kTxQ__, we have provided more description about the secret sharing mechanism.

__The authors would like to kindly remind the reviewers to take a look at the responses and see whether the raised concerns have been well addressed. Thank you for your help and expertise! We look forward to hearing from you again.__

---

### Meta-Review · Program_Chairs · 2022-09-13

**Recommendation:** Accept
**Confidence:** Certain

**Metareview:**

This paper discusses a novel secure aggregation method for dropout resilient federated learning. The proposed solution is interesting, with some reasonable theoretical and empirical analysis. The authors did a good job of engaging with the reviewers in the discussion phase. From my own reading, I found the paper quite lacking in motivation though. The primary motivating scenarios for secure aggregation are for FL from on-premise datasets, such as learning in healthcare settings across hospitals or insurance providers. In such cases, is there a significant dropout concern? On the other hand, I don't see the proposed schemes as practical in the learning from mobile devices scenario, where there is not even a fixed client pool to begin with. It seems that the authors need to think about the motivating scenarios for their work and add discussion about this to the paper.

**Award:**

No

---

### Decision · Program_Chairs · 2022-09-14

Accept